# Monitoring and modeling of lymphocytic leukemia cell bioenergetics reveals decreased ATP synthesis during cell division

Joon Ho Kang [1,2,3,9,10], Georgios Katsikis [1,3,10], Zhaoqi Li[1,3,4], Kiera M. Sapp [1,3,4], Max A. Stockslager[1,3,5], Daniel Lim[1,3], Matthew G. Vander Heiden [1,3,4,6], Michael B. Yaffe [1,3], Scott R. Manalis [1,3,5,7] & Teemu P. Miettinen [1,3,8 ✉]

The energetic demands of a cell are believed to increase during mitosis, but the rates of ATP synthesis and consumption during mitosis have not been quantified. Here, we monitor mitochondrial membrane potential of single lymphocytic leukemia cells and demonstrate that mitochondria hyperpolarize from the G2/M transition until the metaphase-anaphase transition. This hyperpolarization was dependent on cyclin-dependent kinase 1 (CDK1) activity. By using an electrical circuit model of mitochondria, we quantify mitochondrial ATP synthesis rates in mitosis from the single-cell time-dynamics of mitochondrial membrane potential. We find that mitochondrial ATP synthesis decreases by approximately 50% during early mitosis and increases back to G2 levels during cytokinesis. Consistently, ATP levels and ATP synthesis are lower in mitosis than in G2 in synchronized cell populations. Overall, our results provide insights into mitotic bioenergetics and suggest that cell division is not a highly energy demanding process.

[1] Koch Institute for Integrative Cancer Research, Massachusetts Institute of Technology, Cambridge, MA 02139, USA. [2] Department of Physics, Massachusetts Institute of Technology, Cambridge, MA 02139, USA. [3] Center for Precision Cancer Medicine, Massachusetts Institute of Technology, Cambridge, MA 02139, USA. [4] Department of Biology, Massachusetts Institute of Technology, Cambridge, MA 02139, USA. [5] Department of Mechanical Engineering, Massachusetts Institute of Technology, Cambridge, MA 02139, USA. [6] Dana-Farber Cancer Institute, Boston, MA 02115, USA. [7] Department of Biological Engineering, Massachusetts Institute of Technology, Cambridge, MA 02139, USA. [8] Medical Research Council Laboratory for Molecular Cell Biology, University College London, London WC1E 6BT, UK. [9] Present address: Brain Science Institute, Korea Institute of Science and Technology, Seoul 02792, South Korea. [10] These authors contributed equally: Joon Ho Kang, Georgios Katsikis. ✉email: teemu@mit.edu

The process of cell division, which involves widespread reorganization of the cytoskeleton, assembly, and disassembly of mitotic spindles and extensive chromosome movement, is regarded as a high energy-demanding process[1–7]. However, this view that cell division requires large amounts of energy in comparison to processes taking place in interphase has not been directly examined. Although cells contain energy in many forms, herein, we consider adenosine triphosphate (ATP) as the major energy currency used by cells.

In animal cells, including most cancer cells, mitochondrial oxidative phosphorylation is a major source of ATP synthesis[8–10]. Oxidative phosphorylation is carried out by a system analogous to an electrical circuit[11–13], where the electron transport chain (ETC) acts like a battery that couples favorable electron transfer to oxygen to pump protons out of the mitochondrial matrix. This ETC activity creates a mitochondrial membrane potential ($\Delta\Psi$m), i.e., voltage, across the inner mitochondrial membrane. In oxidative phosphorylation, $\Delta\Psi$m is used by ATP synthase to drive the phosphorylation of adenosine diphosphate (ADP). However, protons also leak back into the mitochondrial matrix through other means, thereby decreasing $\Delta\Psi$m without ATP synthesis. Regulation of proton leakage enables cells to uncouple oxygen consumption from ATP synthesis[11–13].

The short duration of mitosis makes studies of cellular bioenergetics (i.e., the changes in ATP levels as well as ATP synthesis and consumption rates) in mitosis challenging. Thus, most studies rely on cell cycle synchronizations and/or prolonged arrests in early mitosis (prometaphase or metaphase). Mitotic arrests are known to result in decreased ATP levels, degradation of mitochondria, activation of AMP-activated protein kinase (AMPK), and increased glycolysis[14,15]. A decrease in mitotic ATP levels has also been suggested by early studies quantifying nucleoside levels in synchronized cells[16,17]. Studies in synchronized cells have also shown that mitochondrial ETC activity increases in mitosis[4,15,18], at least partly due to CDK1 directly activating ETC complex I[4]. Whether this increase in ETC activity is coupled to increased ATP synthesis remains unclear[4,19,20]. Also, the levels of reactive oxygen species (ROS) increase during prolonged mitosis[21]. Overall, these results suggest that the bioenergetic state of cells arrested in mitosis differs radically from interphase cells. Yet, it remains unclear how cellular bioenergetics are altered in normally progressing mitosis, and if mitotic bioenergetics influence mitotic progression.

Recently, single-cell level analyses have begun to reveal how cellular bioenergetics change with the cell cycle in the absence of cell cycle synchronizations[22]. Genetically encoded ATP sensors have revealed that cellular ATP levels gradually decrease from the G2/M transition until the metaphase–anaphase transition, after which ATP levels rapidly recover during the anaphase[3,7,17]. These results have been attributed to the presumably high ATP consumption during early mitosis, although ATP synthesis and consumption in mitosis have not been quantified. In addition, ROS levels are also known to increase during normal mitosis, supporting the notion that the bioenergetic state of cells changes during unperturbed mitosis.

Here, we study mitotic bioenergetics in a lymphocyte model system by combining single-cell high-resolution monitoring of $\Delta\Psi$m with a mathematical modeling approach. This allows us to quantify mitochondrial ATP synthesis rates throughout mitosis without cell cycle synchronizations. We support our findings by measuring bioenergetics in synchronized cell populations and by quantifying mitotic progression in response to bioenergetic perturbations. Surprisingly, our results indicate that mitotic cells reduce their mitochondrial ATP synthesis by approximately 50%, resulting in lowered ATP levels in mitosis. Our work provides a bioenergetic view of mitosis, which argues against the traditional dogma that cell division is a highly energy-demanding process.

## Results

**$\Delta\Psi$m increases transiently prior to cell division.** To study mitochondrial bioenergetics at the single-cell level, we combined suspended microchannel resonators (SMR), a non-invasive single-cell buoyant mass sensor, with a fluorescence-detection system. This allowed us to monitor cell mass-normalized fluorescence signals with a temporal resolution of 2 min and a fluorescence measurement error of 2% without perturbing normal growth[23,24] (Supplementary Fig. 1 and Supplementary Note 1). Using this setup, we examined the murine lymphocytic leukemia cell line L1210 grown in the presence of non-quenching concentrations (10 nM) of tetramethylrhodamine ethyl ester (TMRE), a fluorescent probe for $\Delta\Psi$m[25] (Supplementary Fig. 2a and Supplementary Note 2). Monitoring the mass-normalized TMRE signal over multiple cell generations revealed oscillatory TMRE behavior with a robust, transient, and extensive spike-like increase in TMRE signal preceding the end of each cell cycle (Fig. 1a). As this change in TMRE signal was surprisingly rapid, we validated that the rate of TMRE increase and decrease were not limited by TMRE diffusion speed (Supplementary Figs. 2c–e and Supplementary Note 2). We also validated that the increased TMRE signal at the end of the cell cycle was not specific to our culture conditions, as the TMRE behavior persisted in both glucose and galactose-based culture media (Supplementary Fig. 3a, b). In addition, the TMRE behavior was not cell-type specific as mouse BaF3 pro-B lymphocytes, chicken DT40 lymphoblasts, suspension of HeLa cells, and, importantly, primary CD8+ and CD3+ human T cells also displayed increased TMRE signal at the end of their cell cycle (Supplementary Fig. 3c). However, the TMRE behavior was not universal, as we also found a cell line, mouse Fl5.12 pro-B lymphocytes, that did not display any change in TMRE at the end of the cell cycle (Supplementary Fig. 3c). In all further studies, we utilized L1210 cells as our model system.

First, we wanted to validate that the spike-like TMRE increase reflects an increase in $\Delta\Psi$m. $\Delta\Psi$m-sensitive probes, including TMRE, are also affected by plasma membrane potential ($\Delta\Psi$p), cell volume, and mitochondrial volume (Supplementary Note 2). We used quenching concentrations (10 $\mu$M) of an alternative $\Delta\Psi$m probe, Rhod123 (Supplementary Fig. 2b). In the quenching mode, the fluorescence signal behavior of $\Delta\Psi$m probes is reversed as the probes accumulate in mitochondria at high, quenching concentrations. Rhod123 was not present in the culture media during the experiment, thus making the experiment more specific to $\Delta\Psi$m instead of $\Delta\Psi$p. We found that Rhod123 signal suddenly decreased ~1 h prior to cell division (Fig. 1b and Supplementary Fig. 4a), consistent with a spike-like increase in $\Delta\Psi$m. By contrast, MitoTracker Green, the $\Delta\Psi$m-insensitive probe which reports mitochondrial content, did not display any changes at the end of the cell cycle (Fig. 1c and Supplementary Fig. 4b). We next examined changes in $\Delta\Psi$p using a 1 $\mu$M DiBAC$_4$(3) probe. The DiBAC$_4$(3) signal was reduced before and during the TMRE increase, indicative of increased $\Delta\Psi$p (Supplementary Fig. 4c). We speculated that this $\Delta\Psi$p change might be a feature of mitotic cell swelling[26,27], which can be inhibited with 5-(N-ethyl-N-isopropyl)amiloride (EIPA)[26]. Indeed, the treatment of L1210 cells with 5 $\mu$M EIPA partially inhibited the observed decrease in the DiBAC$_4$(3) signal but did not affect the TMRE signal increase (Supplementary Fig. 4c–e). This indicates that mitotic cell swelling associates with changes in $\Delta\Psi$p. However, the changes in $\Delta\Psi$p and cell volume during the mitotic cell swelling do not significantly affect the reliability of TMRE as a reporter for $\Delta\Psi$m (Supplementary Note 2).

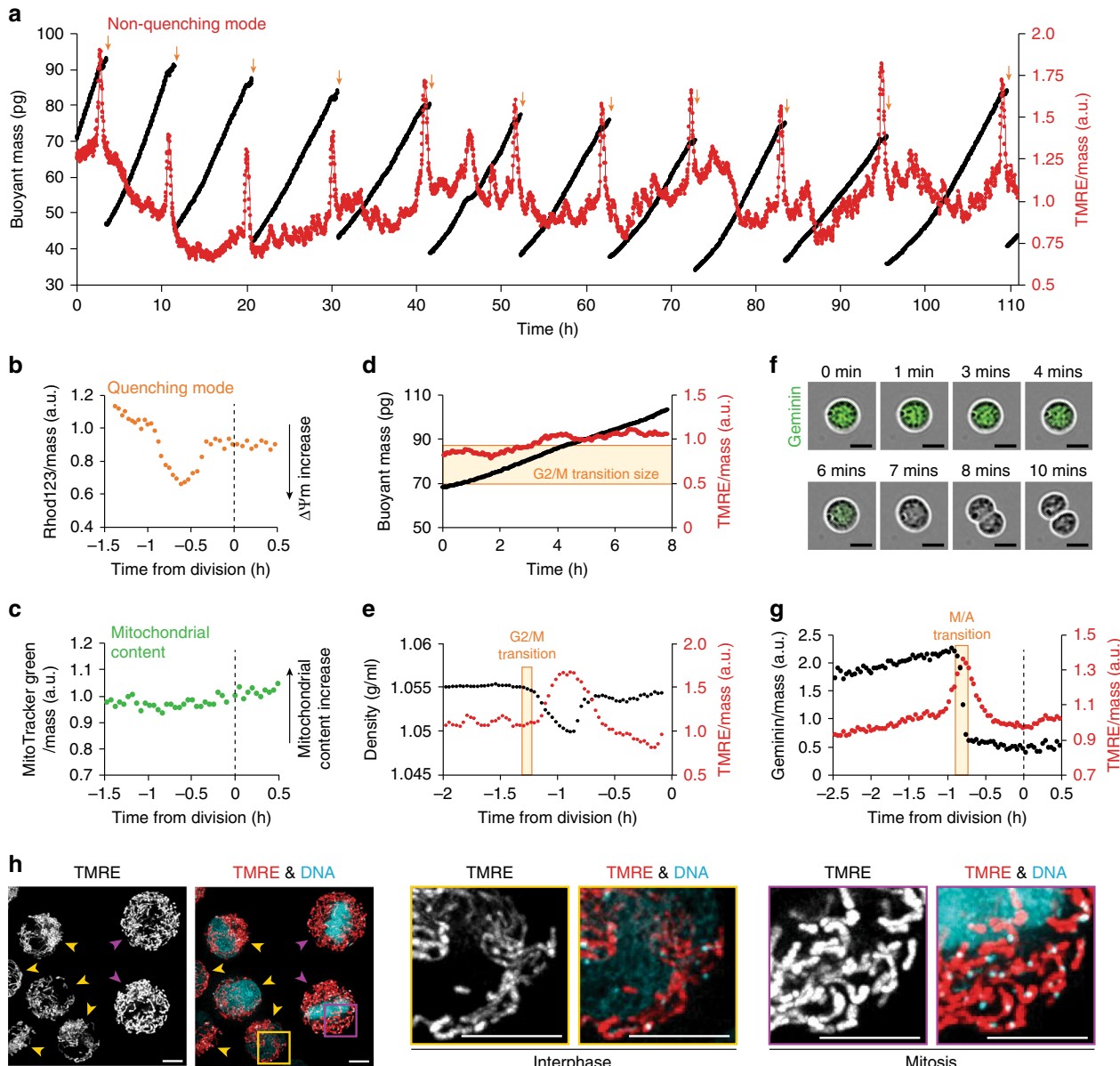

**Fig. 1 Mitochondria transiently hyperpolarize during the prophase and metaphase. a** Buoyant mass (black) and mass-normalized TMRE (red) trace for a single L1210 cell and its progeny over ten full generations with a measurement interval of 1.9 min. At each cell division (orange arrows), one of the daughter cells is randomly kept and monitored, while the other is discarded. TMRE was used in a non-quenching concentration (10 nM). **b** Mass-normalized Rhod132 trace for a L1210 cell around cell division. Cells were loaded with a quenching concentration of Rhod123 (10 µM), washed, and immediately analyzed with no Rhod123 in the culture media. In the quenching mode, the fluorescence signal behavior is reversed. **c** Mass-normalized MitoTracker Green (50 nM) trace for a L1210 cell around cell division. **d** Buoyant mass (black) and mass-normalized TMRE (red) trace for a L1210 cell treated with 2.5 µM RO-3306 to inhibit mitotic entry. The typical size for mitotic entry is illustrated with a light-yellow area. **e** Cell density (black) and mass-normalized TMRE (red) trace for a L1210 cell around cell division. Mitotic entry (G2/M transition) is illustrated with a light-yellow area. **f** Representative phase contrast (gray) and mAG-hGeminin cell cycle reporter (green) images of a L1210 FUCCI cell in mitosis. Experiments were repeated four times independently with similar results. Scale bars: 10 µm. **g** Mass-normalized mAG-hGeminin (black) and mass-normalized TMRE (red) trace for a L1210 FUCCI cell around cell division. Metaphase-to-anaphase transition (M/A transition) is illustrated with light-yellow area. **h** Representative maximum intensity images of live L1210 cells stained with TMRE (red) and PicoGreen (DNA stain, teal). Yellow arrowheads indicate interphase cells, purple arrowheads indicate mitotic cells. Scale bars: 5 µm. Experiments were repeated four times independently with similar results.

**ΔΨm increases in early mitosis and recovers in cytokinesis**. We next studied the exact timing of the spike-like ΔΨm increase. First, we examined if mitochondrial hyperpolarization starts in G2 by stopping cell cycle progression in G2. Inhibition of mitotic entry using 2.5 µM CDK1 inhibitor RO-3306 completely eliminated the observed increase in TMRE signal, despite the fact that cells continued to increase in size beyond the typical G2/M transition[28] (Fig. 1d and Supplementary Fig. 5a). We then

monitored TMRE signal together with biophysical and fluorescent mitotic markers. First, we monitored single-cell density to compare TMRE signal increase to the timing of mitotic cell swelling, an event that is known to start at the beginning of mitosis (prophase)[26,27]. We observed the increase in the TMRE signal immediately following the onset of density reduction, indicating that mitochondrial hyperpolarization begins shortly after mitotic entry (Fig. 1e and Supplementary Fig. 5b). We also

used a cell stiffness-dependent biophysical measurement to mark the G2/M transition and metaphase–anaphase transition[23,24]. This indicated that the TMRE signal starts to increase following the G2/M transition and peaks at the metaphase–anaphase transition (Supplementary Fig. 5c). We next compared the timing of the TMRE signal increase to the degradation of the protein Geminin, which takes place at the metaphase–anaphase transition. Using L1210 FUCCI cells, which express fluorescently labeled Geminin (Geminin-mAG)[24,29], we observed that the Geminin-mAG signal was fully degraded in ~8.6 min at the metaphase–anaphase transition (Fig. 1f and Supplementary Fig. 6). The loss of Geminin-mAG signal coincided with the maximum TMRE signal (Fig. 1g and Supplementary Fig. 5d), thus validating that the TMRE signal peaks at metaphase–anaphase transition. In most cells, the TMRE signal declined back to typical G2 levels before the final abscission of the daughter cells (Fig. 1e, g and Supplementary Fig. 5c, d). We also validated that ΔΨm is higher in mitosis than in G2 using a flow cytometry-based approach. We stained cells with the ΔΨm-sensitive probe MitoTracker Red CMXRos and following fixation, we co-stained the cells for a mitotic marker (p-Histone H3 (Ser10)) and DNA content. Flow-cytometer measurements of these cells revealed that ΔΨm was significantly higher in mitotic cells than in G2 cells (Supplementary Fig. 5e). We reached a similar conclusion when mitotic cells were separated based on Cyclin B levels (Supplementary Fig. 5f). Together, these results indicate that the mitochondrial hyperpolarization begins after the G2/M transition (in prophase), gradually increases during prometaphase and metaphase, reaches a maximum at the metaphase–anaphase transition, and returns to G2 levels during cytokinesis.

**Mitochondria display similar morphologies in interphase and mitosis.** Mitochondrial networks have been reported to fragment during early mitosis in adherent cell types[19,30,31]. Cells can also display extensive variability in mitochondrial morphology, which may influence mitochondrial functionality[32–34]. We examined if mitochondrial morphology and intracellular distribution of TMRE staining vary between mitosis and interphase in our model system. We first imaged active mitochondria using TMRE staining in live L1210 cells. Both mitotic and interphase cells displayed highly connected TMRE-stained mitochondria (Fig. 1h and Supplementary Fig. 7a), suggesting that the morphology of active mitochondria is not radically altered in mitosis. Furthermore, we did not observe differences in the intracellular distribution of the TMRE signal between mitotic and interphase cells (Fig. 1h and Supplementary Fig. 7a), suggesting that mitochondria hyperpolarize uniformly in mitosis, and that the TMRE signal increase in mitosis is not due to increased cytosolic TMRE.

To analyze the mitochondrial morphology independently of ΔΨm, we expressed mitochondrially localized RFP in L1210 cells. In mitosis, the average lengths of mitochondria were lower than in interphase and, consistently, there were more individual mitochondria in mitotic cells than in interphase cells (Supplementary Fig. 7b–d). However, this mitochondrial fragmentation in mitosis was not extensive, as even in mitosis, mitochondria remained mostly connected, and only a few mitochondria covered most of the mitochondrial volume. While the mitotic increase in ΔΨm could still be linked to mitochondrial fission in early mitosis, the decrease in ΔΨm during cytokinesis does not correlate with the known mitochondrial fusion/fission changes[19,30,31].

**CDK1 activity is required for the increase of ΔΨ in early mitosis.** Previous work has shown that the CDK1/Cyclin B complex localizes to mitochondria during mitosis and directly phosphorylates components of the mitochondrial ETC[4]. CDK1 also associates with other metabolic proteins, including the α subunit of ATP synthase[35]. CDK1 activity is minimal before mitotic entry, after which the switch-like activation of CDK1/cyclin B complex results in high CDK1 activity until the onset of anaphase[36–38]. Since the timing of mitochondrial hyperpolarization coincided exactly with the reported CDK1 activity, we hypothesized that the switch-like CDK1 activity is causally responsible for the mitochondrial hyperpolarization. To test this, we first arrested cells in a CDK1 active-state (prometaphase and metaphase) using three different chemicals: the kinesin motor inhibitor S-trityl-l-cysteine (STLC), the microtubule polymerization inhibitor nocodazole, and the anaphase-promoting complex inhibitor proTAME. For all three chemicals, we observed that TMRE signal increased following mitotic entry and plateaued to a high level during the mitotic arrest, indicative of mitochondria reaching a steady, hyperpolarized state (Fig. 2a). The level of mitochondrial hyperpolarization did not depend on the chemical used for mitotic arrest.

Next, we aimed to partially inhibit CDK1 with RO-3306 (1 μM) or with an alternative CDK1 inhibitor BMS-265246 (400 nM) to examine changes in the TMRE signal during STLC-mediated prometaphase arrest. Note that complete inhibition of CDK1 blocks mitotic entry, but it is possible to partially inhibit CDK1 while allowing mitotic entry and progression[24,26]. We validated that RO-3306 and BMS-265246 inhibit CDK1 activity by using western blotting with MPM2 antibody (Fig. 2b), which identifies CDK1/2-phosphorylated sites found on various proteins[39,40]. We also quantified the MPM2 antibody staining using flow cytometry (Fig. 2c). Following partial CDK1 inhibition, we observed lower levels of mitotic mitochondrial hyperpolarization (Fig. 2d, e). We then arrested cells in the prometaphase with STLC and after the TMRE signal had reached a new equilibrium in mitosis we treated the cells with 100 nM okadaic acid (OA). OA inhibits the protein phosphatase PP2A and block the dephosphorylation of CDK1 targets[38], further increasing CDK1 target phosphorylation levels (Fig. 2b–d). The OA treatment increased TMRE signal (Fig. 2f). By contrast, when the CDK1 activity of prometaphase-arrested cells was inhibited with 5 μM RO-3306 the TMRE signal returned to G2 levels (Fig. 2g). Together, these results indicate that CDK1 activity drives the mitochondrial hyperpolarization in early mitosis.

**Mitochondrial ATP synthesis is not required for cell division.** CDK1 has been suggested to promote mitochondrial ATP synthesis[4]. Considering the prevailing dogma that mitosis is energy intensive[1–7], we studied whether acute inhibition of mitochondrial ATP synthesis affected cell division. Direct measurements of oxygen consumption validated that L1210 cells maintain active mitochondrial ATP synthesis, which could be completely inhibited by 1 μM oligomycin, a specific inhibitor of $F_O$-ATP synthase (Supplementary Fig. 8a–c). Unexpectedly, when we treated L1210 cells in the G2 cell cycle phase with 1 μM oligomycin and monitored their growth using the SMR, the cells still proceeded through mitosis and displayed a mild mitochondrial hyperpolarization (Fig. 3a). We quantified the magnitude of the TMRE signal increase in mitosis by arresting cells in mitosis in the presence or absence of oligomycin. This validated that the mitotic mitochondrial hyperpolarization is lower in the presence of oligomycin (Fig. 3b). To further quantitatively analyze the role of mitochondrial ATP synthesis in mitotic entry and progression, we synchronized cells to G2 using RO-3306, treated the cells with 1 μM oligomycin for 15 min, released the cells to enter mitosis in the presence of oligomycin, and collected samples for cell cycle analysis at different timepoints. Surprisingly, mitochondrial ATP

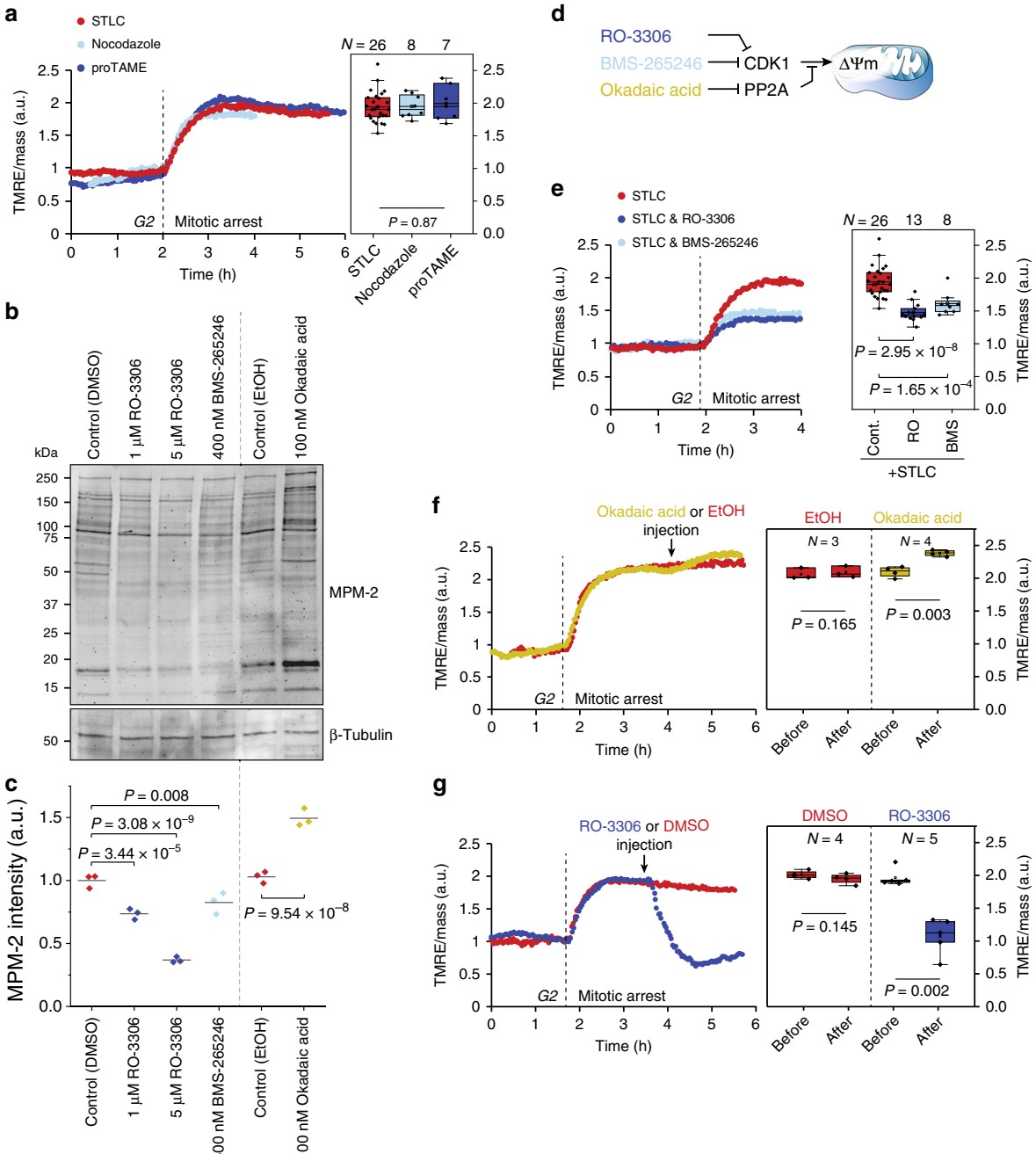

**Fig. 2 CDK1 activity is required for the switch-like mitochondrial hyperpolarization. a** Mass-normalized TMRE traces for L1210 cells treated with 5 µM STLC (dark blue), 1 µg/ml nocodazole (light blue), or 15 µM proTAME (red) to induce mitotic arrest. TMRE signal reaches a stable but highly elevated level when mitosis is prolonged, indicative of a metabolic switch in mitosis. Boxplots show TMRE increases during mitotic arrest. Statistical significance was assessed using one-way ANOVA. **b** Western blot displaying CDK1 target phosphorylation levels using the MPM2 antibody. L1210 cells were treated for 30 min, with indicated chemicals prior to lysis. β-tubulin was used as a loading control. **c** Flow-cytometer-based quantifications of MPM2 antibody labeling. Samples were treated as in panel (**b**). Each dot represents a separate culture ($n = 3$). **d** Schematic indicating how the chemical inhibitors affect CDK1 activity. **e** Mass-normalized TMRE traces for L1210 cells treated with 5 µM STLC alone (red) or in combination with 1 µM RO-3306 (dark blue) or with 400 nM BMS-265246 (light blue) to partly inhibit CDK1. Partial inhibition of CDK1 reduces the extent of the metabolic switch in mitosis. Boxplots show TMRE increases during mitotic arrest. **f** Mass-normalized TMRE traces for a L1210 cell treated with 5 µM STLC. Once the cell was arrested in mitosis, 100 nM okadaic acid (yellow trace) or EtOH (control, 0.1% v/v, red trace) was injected (black arrow) to the culture media. Boxplots show TMRE levels before and after injection. **g** Mass-normalized TMRE trace for a L1210 cell treated with 5 µM STLC. Once the cell was arrested in mitosis, 5 µM RO-3306 (blue trace) or DMSO (control, 0.1% v/v, red trace) was injected (black arrow) to the culture media. Boxplots show TMRE levels before and after injection. Boxplots in (**a**, **e**, **f**, **g**) depict the mean (small square), median (horizontal bar), interquartile range (IQR) (box), and 1.5× IQR (whiskers). In (**a**, **e**, **f**, **g**), N represents independent experiments. Statistical significance was assessed using one-way ANOVA followed by Sidakholm test in (**c**, **e**), or paired, two-tailed Student's t tests (**f**, **g**).

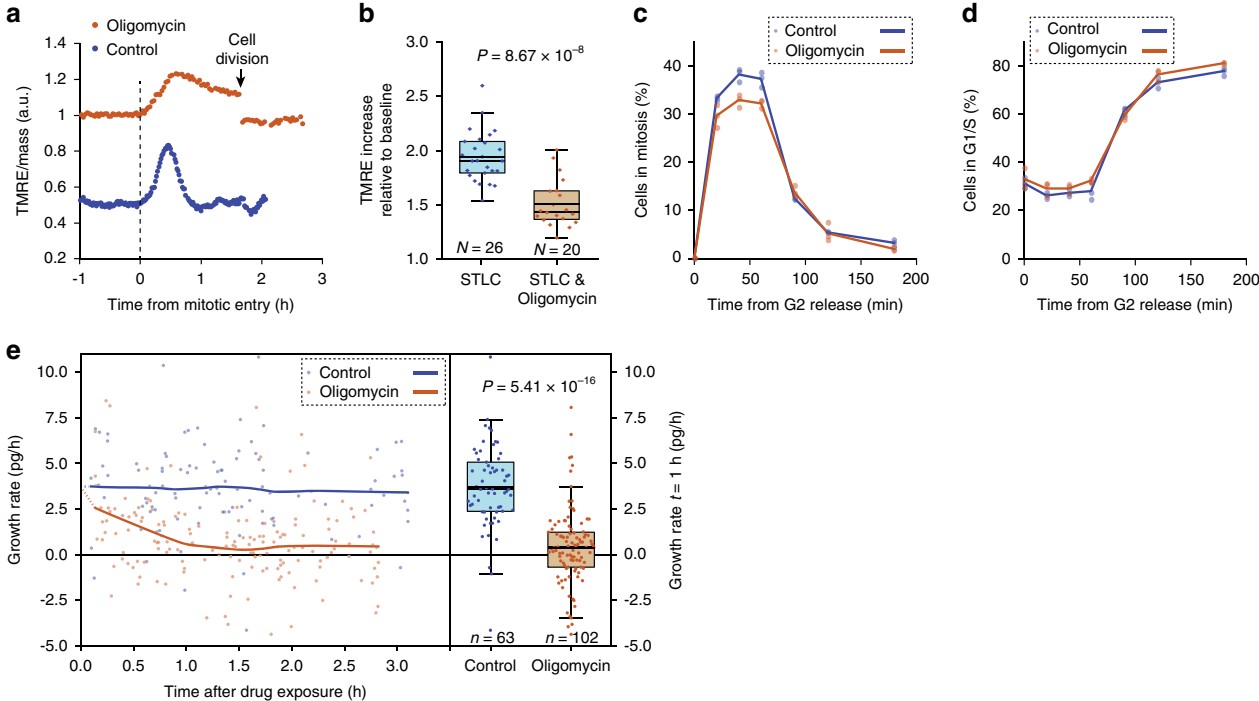

**Fig. 3 Mitochondrial ATP synthase activity is required for cell growth, but not for cell division. a** Mass-normalized TMRE trace for control (blue) and 1 µM oligomycin-treated (brown) L1210 cell around cell division. Both control and oligomycin-treated cells proceed through mitosis, but display distinct TMRE dynamics. **b** Quantifications of the TMRE increase in mitosis following mitotic arrest with STLC in control and 1 µM oligomycin-treated L1210 cells. Baseline refers to G2 TMRE levels. **c** Quantifications of mitotic entry in control (0.1% v/v DMSO-treated) and 1 µM oligomycin-treated L1210 cells. Cells were synchronized to G2, released and collected for cell cycle analysis at indicated timepoints. Treatments were started 15 min before releasing from G2 arrest. Each dot represents a separate culture (n = 3). **d** Quantifications of mitotic exit (appearance of G1 cells) for samples shown in (**c**). **e** Quantification of mass accumulation (growth) rate in individual control and 1 µM oligomycin-treated cells. Each dot represents the growth rate of a single-cell. Quantifications of the growth rates after 1 h drug exposure are shown on the right. In (**b**, **e**), boxplots depict the mean (small square), median (horizontal bar), interquartile range (IQR) (box), and 1.5× IQR (whiskers). Unpaired, two-tailed Student's t test was used for statistical analysis.

synthesis inhibition had little effect on mitotic entry and the subsequent appearance of G1 cells (Fig. 3c, d and Supplementary Fig. 9a). Similar results were observed in BaF3 and DT40 cell lines (Supplementary Fig. 9b–f). To further examine the extent to which ATP synthesis inhibition influences L1210 cell behavior, we monitored single-cell mass accumulation (growth) rates using a serial SMR, which is a high-throughput version of the SMR[24,41]. We observed that oligomycin treatment caused a decrease in cell growth rates that persisted for several hours (Fig. 3e). Thus, mitochondrial ATP synthesis is not acutely required to support cell division, although it does support cell growth. This finding is consistent with prior observations that a mitochondrially localized dominant-negative form of CDK1 did not affect G2/M progression despite reducing mitochondrial respiration[4], and that cells devoid of mitochondrial DNA and mitochondrial ATP synthesis can proliferate, despite significantly reduced growth rates[42].

We also analyzed mitotic entry and progression in the absence of glycolytic activity by releasing G2-arrested L1210 cells into the media lacking glucose. In contrast to mitochondria-specific ATP synthesis inhibition, we observed that cells still entered mitosis but were unable to complete mitosis in the absence of glucose (Supplementary Fig. 9g, h). The rate of mitotic entry in the absence of glucose was slightly reduced by inhibition of mitochondrial ATP synthesis (Supplementary Fig. 9g). Notably, the absence of glucose may influence mitosis through mechanisms other than glycolytic ATP synthesis. Together, these results indicate that mitotic progression has metabolic requirements, but mitochondrial ATP synthesis is not one of these requirements.

**Excess protein expression decreases the ΔΨm increase in early mitosis.** Next, to more directly examine bioenergetics and oxidative stress in mitosis, we expressed various fluorescence-based metabolic reporters in L1210 and BaF3 cells. However, the expression of these exogenous reporter proteins resulted in the loss, or even the reversal of the normal mitotic mitochondrial hyperpolarization (Supplementary Fig. 10a). Moreover, L1210 FUCCI cells displayed lower levels of mitochondrial hyperpolarization than wild-type L1210 cells (Supplementary Fig. 10b, c), despite expressing only low levels of the Geminin-mAG construct. Since lower mitochondrial hyperpolarization in mitosis was observed with all genetic constructs tested, we attribute this change in hyperpolarization to increased protein expression rather than the specific protein that was expressed. As excessive protein expression can perturb the normal cell metabolism in mitosis, these results suggest that the mitochondrial hyperpolarization depends on a minimally perturbed cell state. These results also indicate that the genetically encoded metabolic reporter systems may bias quantitative analyses of mitotic mitochondrial bioenergetics.

**The electrical circuit model reveals mitochondrial ATP synthesis dynamics in mitosis.** To quantify mitochondrial ATP synthesis rates in mitosis without genetically engineering the L1210 cells and without influencing the normal cell cycle progression, we utilized a mathematical model to derive ATP synthesis rates from the time-dynamics of TMRE signal (Supplementary Notes 3, 4 and Supplementary Fig. 11). First, we converted the TMRE signal to approximate ΔΨm (Supplementary

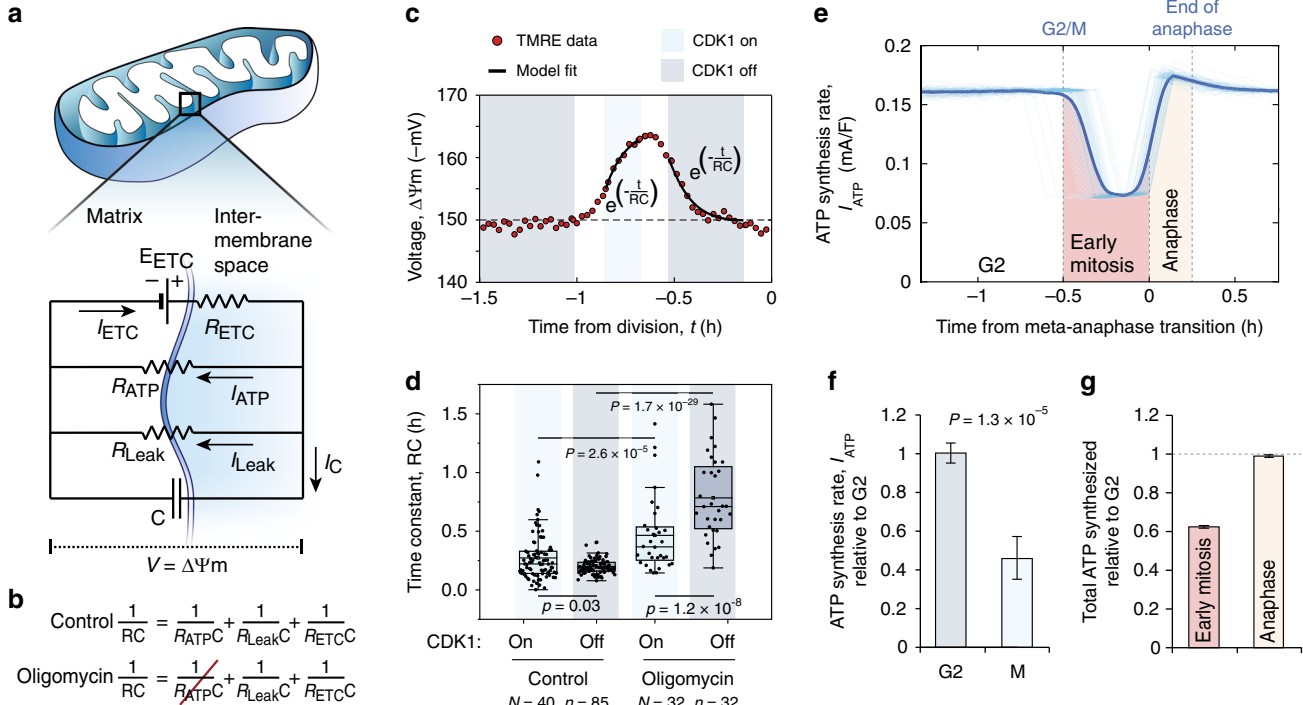

**Fig. 4 Electrical circuit model reveals decreased mitochondrial ATP synthesis in early mitosis. a, b** Electrical circuit model of mitochondria, where $\Delta\Psi$m is the voltage across the circuit (**a**). The key components controlling $\Delta\Psi$m (ETC, ATP synthase, and leakage), all have their respective resistances ($R_{ETC}$, $R_{ATP}$, and $R_{Leak}$), which together form the total resistance ($R$) in the circuit (**b**). Oligomycin increases the $R_{ATP}$ value to infinity. The difference in $R$ values between control and oligomycin-treated cells reflects the $R_{ATP}$ value. **c** Model fit to a typical L1210 single-cell TMRE data. TMRE signals were converted to approximate voltages and CDK1$_{on}$ (light blue) and CDK1$_{off}$ (dark blue) regions were fitted separately to derive the time constant (RC) values for each state. **d** Time constant (RC) values for CDK1$_{on}$ (light blue) and CDK1$_{off}$ (dark blue) regions in control and 1 μM oligomycin-treated L1210 cells. Data depict the mean (small square), median (horizontal bar), interquartile range (IQR) (box), and 1.5× IQR (whiskers). Statistical significance was assessed using one-way ANOVA followed by Sidakholm test. $N$ denotes the number of independent experiments, $n$ denotes the number of cells. **e** ATP synthesis rate ($I_{ATP}$) for 85 separate L1210 cells around cell division. Individual cells are drawn with thin opaque lines; the population average is drawn with a thick solid line. The colored areas reflect total amounts of ATP synthesized during early mitosis (red) and anaphase (light yellow). **f** Quantifications of relative ATP synthesis rates ($I_{ATP}$) in G2 (dark blue) and mitosis (light blue) as modeled on a single-cell level in non-arrested cells. G2 rates were obtained using RC values for CDK1 off, and the $\Delta\Psi$m observed prior to mitotic entry. Unpaired, two-tailed Student's $t$ test was used for statistical analysis. **g** Quantifications of the total ATP synthesized by mitochondria during early mitosis (from G2/M to M/A transition, red) and during anaphase (light yellow), when compared to a null-hypothesis where ATP synthesis rate remains at G2 levels throughout mitosis (dashed horizontal line). For panels (**f, g**), data depict mean ± SEM (see Supplementary Note 8 for error analysis), and the data are the same as in panel (**d**).

Note 5)[43]. Second, we assumed that the voltage across the inner mitochondrial membrane ($\Delta\Psi$m) is determined by the currents through ATP synthesis ($I_{ATP}$), proton leak across the membrane ($I_{Leak}$), and the ETC ($I_{ETC}$). Third, as in previously published and biochemically relevant models[13,25], we considered the inner mitochondrial membrane as an electrical circuit with voltage ($\Delta\Psi$m), capacitance (C), and resistances (R) for each one of the currents (Fig. 4a and Supplementary Note 3). Because we found that CDK1 activity is responsible for the mitochondrial hyper-polarization (Fig. 2), and because CDK1 has been shown to activate in a switch-like manner[36–38], we set the circuit to behave in a switch-like manner between active CDK1 (CDK1$_{on}$, from the G2/M transition to the metaphase–anaphase transition) and inactive CDK1 (CDK1$_{off}$, G2 and cytokinesis) states, while accounting for the short duration when CDK1 activity is converting from one state to another (Supplementary Note 5). By fitting our model's analytical solution to the $\Delta\Psi$m data, we derived RC values, which reflect the time constants of the $\Delta\Psi$m change, for each control and oligomycin-treated cell during CDK1$_{on}$ and CDK1$_{off}$ states (Fig. 4b–d). Comparing the RC values between control and oligomycin-treated cells, we extracted the resistance of ATP synthase ($R_{ATP}$) during the CDK1$_{on}$ and CDK1$_{off}$ states (Fig. 4b). We found that $R_{ATP}$ is higher during the

CDK1$_{on}$ state than during CDK1$_{off}$ state (Supplementary Fig. 12c and Supplementary Note 3). In addition, comparing RC values during the CDK1$_{on}$ and CDK1$_{off}$ states in each control cell revealed that in order for $R_{ATP}$ to increase, $R_{Leak}$, and $R_{ETC}$ are required to cumulatively decrease during the CDK1$_{on}$ state (Supplementary Note 6), suggesting increased ETC activity and/or increased proton leakage in mitosis. We then derived the current through ATP synthase ($I_{ATP}$), i.e., the ATP synthesis rate, throughout mitosis using Ohm's law ($I_{ATP} = V/R_{ATP}$) (Supplementary Notes 6–8). We also utilized the CDK1$_{off}$ state RC values to derive ATP synthesis in late G2. Surprisingly, our modeling revealed that the mitochondrial ATP synthesis rate is decreased by 54% ± 11% (mean ± SEM) during prometaphase and metaphase, compared to ATP synthesis rates in G2 (Fig. 4e, f). During anaphase, only a minor increase (<10%) in ATP synthesis relative to G2 levels was observed (Fig. 4e). Overall, this temporal control of ATP synthesis results in 40% decrease in the total ATP synthesized by mitochondria during early mitosis (between G2/M transition and metaphase–anaphase transition) when compared to a situation where mitochondrial ATP synthesis would remain at G2 levels (Fig. 4g).

It is important to recognize the limitations of our approach. In particular, (i) the TMRE signal is a proxy for $\Delta\Psi$m, and subject to

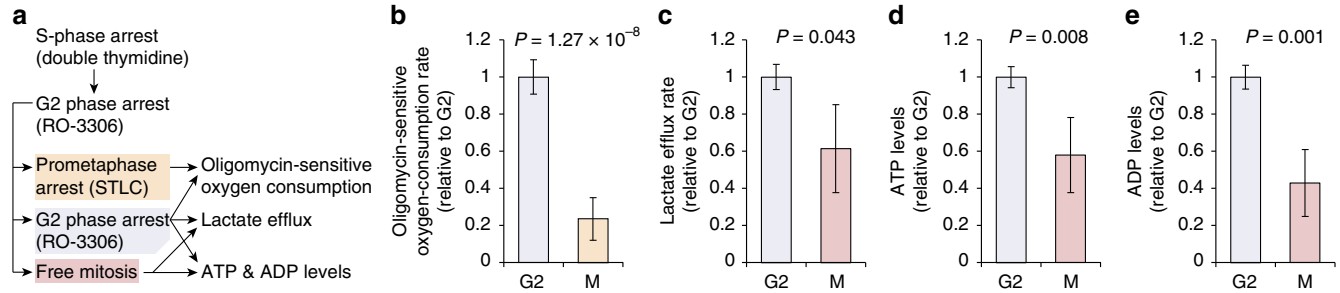

**Fig. 5 Flux measurements in synchronized cell populations reveal decreased ATP synthesis and ATP levels in mitosis relative to G2. a** Schematic of cell cycle synchronization procedure. Oxygen-consumption rates were measured using cells arrested in mitosis due to the time needed to obtain measurements. Lactate efflux and ATP and ADP levels were measured in cells progressing through mitosis. All results were normalized by the number of cells in mitosis and G2. **b** Oligomycin-sensitive oxygen-consumption rate, i.e., mitochondrial ATP synthesis rate, in G2 and mitotic cells ($N = 7$–8 independent synchronizations and cultures). **c** Lactate efflux rate in G2 and mitotic cells ($N = 4$ independent synchronizations and cultures). **d** ATP levels in G2 and mitotic cells ($N = 5$ independent synchronizations and cultures). **e** ADP levels in G2 and mitotic cells ($N = 5$ independent synchronizations and cultures). For panels (**b**–**e**), data depict mean ± SD (compound errors after accounting the mean values for imperfect cell cycle synchrony). Unpaired, two-tailed Student's $t$ test was used for statistical analysis.

systematic errors, (ii) we assumed that oligomycin only perturbs mitochondrial ATP synthase, and (iii) our electric circuit model, while including all major elements of mitochondrial bioenergetics, may oversimplify the dynamics of ATP synthesis. However, we observed that the conclusions about ATP synthesis have little sensitivity to the model-specific parameters, including those used for TMRE-to-ΔΨm conversion (Supplementary Figs. 13–15 and Supplementary Note 9).

**ATP levels and synthesis are decreased in synchronized cells in mitosis.** To support our modeling results with a more comprehensive view of bioenergetics in mitosis, we measured metabolic fluxes in L1210 populations synchronized in G2 or in mitosis (Fig. 5a). We first quantified the average ATP synthesis rates of prometaphase and G2-arrested cell populations by measuring oligomycin-sensitive oxygen consumption. Consistent with our modeling approach, the reduced oligomycin-sensitive oxygen-consumption rates in mitotic cells indicated that the mitochondrial ATP synthesis rate is decreased in mitotic cells when compared to G2 cells (Fig. 5b). Furthermore, oxygen-consumption measurements revealed that cells arrested in mitosis have higher respiration in the presence of oligomycin, indicative of increased proton leakage in mitotic cells when compared to G2 cells (Supplementary Fig. 8d, e). This is also consistent with our modeling.

It is possible that the decrease in mitochondrial ATP synthesis in mitosis could be compensated for by increased glycolytic ATP synthesis. To examine the rate of glycolysis, we synchronized L1210 cells to G2 and compared the lactate secretion rate from cells arrested in G2 with that observed in cells released into mitosis (Fig. 5a). Surprisingly, this revealed that the lactate efflux rate is also decreased in mitosis when compared to G2-arrested cells (Fig. 5c). Finally, we also examined cellular ATP and ADP levels in the same synchronized cells. Both ATP and ADP levels were lower in mitotic cells compared to G2 cells (Fig. 5d, e). This is consistent with decreased ATP synthesis in mitosis, as well as previous reports regarding mitotic ATP levels[3,7,16,17].

## Discussion

Overall, our work reveals previously unknown time-dynamics of ΔΨm and mitochondrial ATP synthesis during mitosis. We were able to combine high-resolution monitoring of ΔΨm (Fig. 1a) and chemical perturbations of the mitochondria (Fig. 3a) with mathematical modeling (Fig. 4a, b) to derive mitochondrial ATP synthesis rates in mitosis for a mouse lymphocytic leukemia cell line (Fig. 4e). This revealed that mitochondrial ATP synthesis is inhibited by 50% during early mitosis. We confirmed this result using population-based measurements of bioenergetics, which suggested that mitotic cells have lower ATP synthesis rates as well as ATP and ADP levels in comparison to G2 cells (Fig. 5). We have validated the changes in ΔΨm using multiple approaches (Fig. 1 and Supplementary Fig. 5e). Importantly, we also observed the mitotic ΔΨm changes in other suspension grown cells, including primary T cells, but not in the Fl5.12 pro-B lymphocyte cell line (Supplementary Fig. 3), indicating that the ΔΨm change is not a requirement for cell division. Furthermore, the expression of exogenous proteins influenced the mitotic ΔΨm behavior so that we could only observe the mitotic ΔΨm increase in minimally perturbed cells (Supplementary Fig. 10). Mechanistically, the mitotic changes in ΔΨm are at least partly driven by CDK1 activity (Fig. 2), but further work will be needed to decipher the detailed molecular mechanism behind this. While CDK1 can interact with ATP synthase[35], the inhibition of ATP synthesis could also be influenced by, for example, changes in mitochondrial morphology. Importantly, the mechanism through which ATP synthase is inhibited does not influence the conclusions of our electrical circuit modeling.

Our findings are consistent with existing literature. Increased ΔΨm at the end of the cell cycle has been reported previously[4,44], although previous approaches have not been able to reveal the dynamics of ΔΨm. The observation that ΔΨm can increase in mitosis in the presence of oligomycin (Fig. 3a, b) is consistent with the reported mitotic activation of the ETC[4,15,18]. However, it is important to note that ETC activity can be uncoupled from mitochondrial ATP synthesis, and this coupling has not been examined by previous studies measuring mitotic ETC activity. Indeed, our results suggest that ΔΨm increases due to both ETC activation and ATP synthesis inhibition. Increased ΔΨm is known to promote mitochondrial protein import[18], ROS generation[45], proton leakage[46], and heat production[47]. Consistently, mitochondrial protein import, ROS levels, and cellular heat output have been reported to increase during mitosis in a CDK1-dependent manner[18,21,40,48]. Finally, the mitochondrial ATP synthesis dynamics we describe (Fig. 4e) are fully consistent with the reported dynamics in cellular ATP levels in mitosis[3,7,17].

Our work provides an approach for quantifying mitochondrial ATP synthesis through single-cell ΔΨm monitoring and modeling. This single-cell approach provided similar results to those obtained with traditional, population-based oxygen-consumption measurements (Figs. 4f, 5b). While the latter is easier to carry out

and more widely applicable to different biological situations and cell types than the former, the use of both approaches substantiates the conclusions. Unlike population-based oxygen-consumption measurements, our single-cell monitoring and modeling approach also allows us to quantify the dynamics of mitochondrial ATP synthesis rates, which in turn enables calculation of the total ATP synthesized by mitochondria in mitosis (Fig. 4g). Importantly, our approach achieves this in freely proliferating cells without any cell cycle synchronizations, thereby removing synchronization induced biases present in population-based measurements.

How energy consuming is the process of cell division? Cell division is often assumed to consume large amounts of ATP due to the extensive cellular remodeling taking place but, to our knowledge, this has never been quantitatively examined. We can provide ballpark estimates of ATP consumption in mitosis from these results. Our modeling results (Fig. 4f) and population-based bioenergetic measurements (Fig. 5b, c) suggest that overall cellular ATP synthesis is decreased by ~50% in mitosis when compared to G2. Similarly, ATP levels decreased ~40% in mitosis (Fig. 5d). In interphase, ATP consumption is estimated to consume cellular ATP within approximately a minute[40]. Thus, for the duration of mitosis (~30 min) the rate of ATP consumption must decrease by nearly as much as the observed ~50% decrease in the rate of ATP synthesis. However, this number should be considered only as a rough estimate, as (i) population-based bioenergetic measurements rely on cell cycle synchronizations, which can influence cell growth and metabolism[24,49], (ii) estimation of glycolytic ATP synthesis relies on measurements of lactate output, but glycolytic flux may be directed to different metabolic pathways in mitosis, including lipid synthesis and pentose phosphate pathways[50,51], and (iii) bioenergetics in mitosis may depend on the cell type, and our bioenergetics measurements come from a single-cell type. Nonetheless, our results still imply that the rate of ATP consumption is lower in mitosis than in interphase. This argues against the traditional view that cell division requires high amounts of energy in comparison to interphase. Indeed, if mitosis did require significantly more ATP than interphase, one would expect that inhibition of ATP synthesis would prevent cell division. However, we did not observe this when mitochondrial ATP synthesis was acutely inhibited (Fig. 3c, d).

For fast-growing cancer cells, such as L1210s, macromolecule synthesis processes are considered to be responsible for a large fraction of cellular ATP consumption[52,53]. In mitosis, RNA synthesis will be decreased due to the chromatin condensation, disassociation of transcription factors from DNA and disappearance of nucleoli[54,55]. However, in L1210 cells mitotic protein synthesis rates are similar to those observed in interphase[24], thereby consuming significant amounts of ATP. As the overall ATP consumption still decreases, the events specific for cell division, such as cellular reorganization, are likely to be relatively minor consumers of ATP.

Cells maintain ATP concentrations in the low millimolar range[7,56], but most enzymes in the cellular environment are believed to have Michaelis constants ($K_m$) for ATP in the micromolar range[57]. Thus, even a relatively large decrease in cellular ATP levels, including the ~40% decrease that we observe between mitotic and G2 cells, may not limit enzymatic reaction rates, although allosteric regulation of enzymes by ATP could be affected. Importantly, ADP levels also decreased in mitosis, resulting in an approximately stable ATP/ADP ratio. Mitochondrial ATP synthase activity is sensitive to ATP and ADP levels, and it remains possible that decreased ATP consumption is partly responsible for the ATP synthase inhibition in mitosis. However, our results also suggest that CDK1 has a direct or indirect role in

regulating ATP synthesis. Finally, the decreased ATP and ADP levels could result in increased AMP levels. Indeed, AMP-activated protein kinase (AMPK) activation has been reported to take place in and even promote mitosis[58,59], although not all studies have observed this[15].

The decreased ATP levels could influence mitosis in a manner independent of ATP's role in bioenergetics. ATP binds $Mg^{2+}$ and thereby reduces the amount of free $Mg^{2+}$ in a cell[7,60]. Consistently, the decrease in cellular ATP levels in mitosis has been shown to result in increased levels of free $Mg^{2+}$, which in turn promotes chromatin condensation[7]. Therefore, the mitotic inhibition of ATP synthesis and the resulting low ATP levels may promote specific processes during cell division. Furthermore, ATP is known to act as a hydrotrope that solubilizes membraneless organelles and disordered proteins[57,61]. Curiously, these ATP-sensitive proteins have been shown to be stabilized and less soluble in mitosis[62]. Thus, our finding of reduced mitotic ATP synthesis, and consequently reduced ATP levels, could function as a mechanism to stabilize disordered proteins in mitosis. Importantly, increasing protein solubility in mitosis has been shown to disassemble the mitotic spindle[62]. Thus, reducing ATP levels in mitosis by reducing ATP synthesis may promote successful cell division.

## Methods

**Cells and cell culture conditions**. L1210, BaF3, DT40, Fl5.12, and primary T cells were cultured in RPMI that contained 10% FBS (Gibco), 11 mM glucose, 2 mM glutamine, 1 mM Na pyruvate, 20 mM HEPES, and antibiotic/antimycotic. In addition, Fl5.12 culture media was supplemented with 10 ng/ml IL-3 (R&D Systems), DT40 culture media was supplemented with 3% chicken serum (Sigma-Aldrich) and primary T-cell culture media was supplemented with 10 mM 2-mercaptoethanol, 100 U/ml IL-2 (R&D Systems) and 2 mg/ml anti-human CD28 (BioLegend). S-HeLa cells were grown in DMEM that was supplemented with 10% FBS, 1 mM sodium pyruvate, and antibiotic/antimycotic. L1210 cells were also grown in modified RPMI when comparing growth conditions (Supplementary Fig. 3a, b). For these experiments, we used RPMI containing 4% FBS and either high (25 mM) glucose, low (4 mM) glucose, or 10 mM galactose with no glucose.

Isolation of naive CD3+ and CD8+ primary T cells was carried out from an unpurified buffy coat (Research Blood Components) from which PBMCs were isolated using Ficoll-Paque Plus density gradient (GE). After isolation of the PBMCs, the cells were subjected to red blood cell lysis using ACK lysis buffer (Thermo Fisher Scientific). The PBMCs were then washed three times, and T cells were isolated using Naive CD3+ or CD8+ T Cell Isolation Kit (Miltenyi Biotec), according to kit instructions. The cells were then activated by culturing the cells on an anti-CD3 coated cell culture plate in the media detailed above. T cells were used for experiments approximately 30 h after activation. The activation of T cells was validated by monitoring cell volumes and counts using coulter counter (Beckman Coulter).

Within all SMR experiments, the culture media used was identical to that described above, apart from any indicated fluorescent probe and/or chemical inhibitor addition. All cell culture reagents were obtained from Invitrogen, unless otherwise stated. L1210 cells we obtained from ATCC (Cat# CCL-219), BaF3 cells were obtained from RIKEN BioResource Center (Cat# RCB4476), DT40 cells, which harbor a CDK1as[63], were a gracious gift from K. Samejima and B. Earnshaw from University of Edinburgh, Fl5.12 cells were a gracious gift from M. Vander Heiden from Massachusetts Institute of Technology, and S-HeLa cells were a gracious gift from K. Elias from Brigham Women's Hospital. All cell lines were tested to be free of mycoplasma.

**Generation of reporter cell lines**. The FUCCI cell cycle marker expressing L1210 cells were generated in a previous study[29]. Cells that stably express an ATP reporter (A-Team), reactive oxygen species reporter (roGFP2-Orp1), glutathione redox potential reporter (Grx1-roGFP2) or mitochondria localized RFP (mito-RFP) were generated using lentiviral vectors obtained from AddGene[64–68]. pEIGW roGFP2-Orp1 was a gift from Tobias Dick (Addgene plasmid #64993; http://n2t.net/addgene:64993; RRID:Addgene_64993), pEIGW Grx1-roGFP2 was a gift from Tobias Dick (Addgene plasmid # 64990; http://n2t.net/addgene:64990; RRID:Addgene_64990), ATeam1.03-nD/nA/pcDNA3 was a gift from Takeharu Nagai (Addgene plasmid # 51958; http://n2t.net/addgene:51958; RRID:Addgene_51958), pclbw-mitoTagRFP was a gift from David Chan (Addgene plasmid # 58425; http://n2t.net/addgene:58425; RRID:Addgene_58425). Lentiviruses for the genetically encoded sensors were produced by calcium phosphate transfection of plasmids into HEK-293T cells. For each sensor, the corresponding viral vector was co-transfected with separate plasmids expressing VSV-G and Gag/Pol. Following a media change after an overnight transfection, virus aliquots were harvested at 48 h and 72 h post

transfection and filtered through 0.45-μm filters. Lentiviruses containing Ca$^{2+}$ reporter (GCaMP3)[69] vectors were obtained from Kerafast (cat# FCT188). All metabolic sensors were expressed under strong promoters, but FUCCI cell cycle reporter was expressed under the weak endogenous Geminin promoter.

Lentiviruses were transfected into L1210 and BaF3 cells using spinoculation. Approximately $1.5 \times 10^5$ cells were mixed with 10 mg/ml polybrene (EMD Millipore) and the lentiviruses. This mixture was centrifuged at 800 $g$ for 60 min at 25 °C, after which the cells were moved to normal culture media for 12 h. This was repeated four times. Twenty-four hours later, the selection process was started with puromycin or neomycin, depending on the transfected vector. Following 5 days of selection, a subpopulation with high fluorescence reporter expression was sorted out using BD FACS Aria.

**Suspended microchannel resonator (SMR) setup.** The SMR devices were fabricated at CEA-LETI, France. The exact dimension and geometry of the device can be found in ref. [23]. The SMR devices were actuated the second vibration mode by a piezo-ceramic placed underneath the chip. The SMR devices were operated in a closed-loop, where the output motion of the cantilever is amplified, delayed, and fed back to the piezo-ceramic to drive the cantilever. The signal delay was chosen so that the amplitude of the cantilever motion was maximized. The amplification was set to minimize the resonant frequency error, but not in the limit where it distorts the linearization of the resonant frequency. A digital platform was implemented to track the change in resonant frequency (frequency of the closed-loop signal) over time. This closed-loop operation configuration results in measurement bandwidth ~1500 Hz, wide enough to capture the frequency modulation during the cell transit event, which is typically set to ~150–200 ms. The resonance frequency changes were analyzed and converted to buoyant mass as detailed in refs. [23,28]. The serial SMR experiments (Fig. 3e) were carried out as detailed previously[24,41,70].

**Optical detection setup for SMR.** The optical setup used in this study is similar to our previous setup[23], except a few additions/modifications in optical components to achieve simultaneous measurements of TMRE and FUCCI signals. First, we implemented a "full multiband configuration", which is composed of a multiband exciter filter (Semrock, FF01- 387/485/559/649-25), a multiband emission filter (Semrock, FF01- 440/521/607/694/809-25), and a multiband dichroic beam splitter (Semrock, FF 408/504/581/667/762-Di01-25×36). This filter set cube was mounted in our microscope (Nikon). Second, a dichroic beam splitter (Semrock, FF580-FDi01-25×36) was mounted beyond the image plane to separate the multi-color fluorescence output signal into single-color fluorescent signals (i.e., FUCCI and TMRE signals). The split fluorescent signals were measured by photomultiplier tubes (Hamamatsu, H10722-20) mounted on top of the microscope with single band-pass filters (Semrock, FF01-520/35 for FUCCI, Semrock, FF01-607/35 for TMRE, respectively). Third, to minimize phototoxicity from long-term measurements, the excitation light source was kept on for <500 ms for each measurement, during which the cell was flown through the excitation light path. The cell was directly exposed to the light for ~50 ms during each measurement. This fluorescence measurement was carried out approximately every 2 min (in conjugation with every second buoyant mass measurement) (Supplementary Fig. 1a, b). The area of light exposure was limited 60 × 60 μm, and the area of emission collection was limited to 40 × 60 μm. We did not observe changes to single-cell growth rates when measuring TMRE signal with this setup. All parts of the optical detection setup (including objective lens: 50×/0.55-NA Nikon-CFI, LU Plan ELWD WD 10.1 mm; rectangular slit for emission area control: Thorlab; light source: Lumencor, Spectra X Light Engine) were identical to those used before[23]. The optical path for simultaneous two-color measurements is similar to what has been shown in the previous work[29].

**System operation and error quantification.** Buoyant mass and fluorescence signals were continuously measured using previously explained hydrodynamic trapping approach[23,24,28,29]. The hydrodynamic trapping, SMR measurements, and optical operation were controlled using LabVIEW 2012 software and are explained in more detail in Supplementary Fig. 1a, b.

We quantified the error in optical measurement by repeatedly measuring fluorescently labeled fixed cells. We then carried out linear fitting to the data and calculated the average deviation of each measurement from a linear fitting. The signal-to-noise ratio displayed in Supplementary Fig. 1c, d was calculated by dividing the mean fluorescence intensity of each cell by the standard deviation of the measurements from the fitted line. Using this approach, we estimated our optical measurements to have an error (CV) of approximately 2% for mitotic cells. See Supplementary Fig. 1c, d for details. The error in buoyant mass measurement was quantified previously (by repeatedly measuring a 10-μm polystyrene bead) to be <0.1 pg, which corresponds to an error of <0.2%[24].

**Detecting cell cycle transitions within the SMR.** Three cell cycle transitions were assigned for each cell in the SMR data. First, the G2/M transition was detected using the node deviation signal (Supplementary Fig. 5c), which is an acoustic, stiffness-dependent signal detected by the SMR[23]. This timing was also validated by measurements of mitotic cell swelling (Fig. 1e and Supplementary Fig. 5b), which

start immediately following mitotic entry[26,27]. Single-cell density was measured as detailed in ref. [26]. Second, the metaphase–anaphase transition was detected using the node deviation signal (Supplementary Fig. 5c) and also using the rapid change in buoyant mass accumulation rates[24]. The timing of metaphase–anaphase transition was further validated using the FUCCI cell cycle sensor expressing cells (Fig. 1f, g and Supplementary Fig. 5d). The imaging of FUCCI cells (Fig. 1f) was carried out using IncuCyte. Third, the abscission of the daughter cells was assigned using the sudden ~50% reduction in buoyant mass (Fig. 1a). In addition, we assigned anaphase to the last 15 min (Fig. 4e), as based on our previous quantifications of L1210 cell-elongation duration[23].

**Membrane potential and mitochondrial content measurements.** For all cells, mitochondrial membrane potential was examined using TMRE (Invitrogen) in 10 nM (non-quenching) concentration, with the exception that DT40 cells and primary T cells were measured using 20 nM TMRE. TMRE was detected on the TRICT channel of our SMR setup (see above). L1210 cells at a confluency of ~300.000 cells/ml were stained with 10 nM TMRE for 30 min, after which cells were loaded into the SMR, where the cells were grown in the presence of 10 nM TMRE. Mitochondrial membrane potential was also measured using 10 μM (quenching) concentrations of the Rhod123 (Invitrogen). Rhod123 was detected on the FITC channel of our SMR setup (see above). Cells were stained for 45 min with 10 μM Rhod123, washed twice with media and loaded into the SMR. No Rhod123 was present in the media within the SMR. Note that Rhod123 slowly diffuses out of the cell and this excludes long-term experiments. In order to observe mitosis before Rhod123 diffused out of the cell, we carried out Rho123 experiments by loading only large G2 cells into the SMR. For both TMRE and Rhod123, the quenching and non-quenching states were validated by staining L1210 cells as described above and quantifying fluorescence levels after 30 min of treatment with 1 μM FCCP or oligomycin using BD Biosciences flow cytometer LSR II HTS with excitation lasers at 488 nm and 561 nm, and emission filters at 530/30 and 585/15.

When analyzing the error of mitochondrial fluorescence measurements, L1210 cells were stained for 30 min with MitoTracker Red CMXRos (Invitrogen), after which cells were washed with PBS and fixed in 4% PFA for 10 min. The cells were then washed, and the mitochondrial signal of each cell was repeatedly measured in SMR using the TRICT channel of our SMR setup (see above).

Mitochondrial content was examined using the MitoTracker Green probe (Invitrogen) in 50 nM concentration. MitoTracker Green was detected on the FITC channel of our SMR setup (see above). As with TMRE, cells were stained for 30 min after which cells were loaded into SMR. MitoTracker Green was present in the culture media within SMR. Note that in these concentrations, the presence of TMRE and MitoTracker Green did not affect cell growth rate.

Plasma membrane potential was examined using Bis-(1,3-dibutylbarbituric acid) trimethine oxonol (DiBAC4(3)) probe (Invitrogen) in 1 μM concentration. DiBAC4(3) was detected on the FITC channel of our SMR setup (see above). Cells were stained for 45 min, after which cells were loaded into the SMR. Within the SMR, cells were grown in the presence of DiBAC4(3).

When using a flow cytometer to measure live cell mitochondrial or plasma membrane potential, TMRE and DiBAC4(3) were used at 10 nM and 1 μM concentrations, respectively, and cells were analyzed in the presence of the fluorescence probes. When using a flow cytometer to measure mitochondrial membrane potential changes between G2 and mitosis, L1210 cells were stained for 30 min with 100 nM MitoTracker Red CMXRos (Invitrogen), after which cells were washed with PBS and fixed in 4% PFA for 10 min. The fixed cells were stained for DNA and for p-Histone H3 or Cyclin B as detailed below ("cell cycle synchronizations") and in ref. [24]. The cell cycle marker and MitoTracker Red CMXRos labeling were quantified using BD Biosciences flow cytometer LSR II HTS. See Supplementary Fig. 5e, f, for example, FACS plots with gating.

**Mitochondrial imaging and image analysis.** For all imaging experiments, L1210 cells were grown under identical conditions to those used in SMR experiments. The cells were stained with PicoGreen DNA dye (Thermo Fischer Scientific) for 1 h under normal growth conditions. Note that PicoGreen stains both nuclear and mitochondrial DNA. When imaging TMRE-stained mitochondria, cells were stained for an additional 30 min with 20 nM TMRE. Then, the cells were washed twice with culture media and plated on coverslips coated with 0.1% poly-L-lysine. For TMRE imaging, 20 nM TMRE was maintained in the media during plating and imaging. Imaging was carried out at 37 °C and started 30 min after plating. Imaging was carried out using a DeltaVision wide-field deconvolution microscope with standard filters (FITC, TRITC) and ×100 oil-immersion objective. We utilized immersion oil with a refractive index of 1.516 (Cargille Laboratories). For each cell, we collected ~75 z layers, each 0.2-μm apart. The image xy resolution was ~0.1 μm, and z resolution 0.2 μm. Ten rounds of image deconvolution were carried out using SoftWoRx 7.0.0 software.

Images were analyzed using ImageJ (version 2.0.0-rc-69/1.52p). Each cell was cropped out into its own image stack, and the top and bottom z layers which did not contain the cell were removed. The TRITC channel with mito-RPF signal was converted to 8-bit grayscale image and then converted to a binary image based on a user-defined threshold value. We extracted the pixel intensity for each image stack and observed that mitotic cells displayed lower total intensities. To avoid bias in mitochondrial network morphology analysis, the threshold value used for

conversion to the binary image was based on a visual impression for reproducing the mitochondrial network. To extract the three-dimensional mitochondrial network across the z stack from the binary images, the *Skeletonize (2D/3D)* and *Analyze Skeleton (2D/3D)* functions of ImageJ were used. Out of every image stack, the total length of the three-dimensional mitochondrial network along with the average length and the total number of mitochondrial structures were extracted.

**Chemical perturbations.** Chemical treatments within the SMR were carried out by adding the chemical of interest to the culture media within the SMR. Thus, cells were exposed to the chemical when loading the cells into SMR, which in a typical experiment was approximately 2 h prior to mitosis. Where separately indicated (Fig. 2f, g), chemicals were injected into the SMR culture media during the experiment. All SMR experiments with chemical treatments were stopped following mitotic exit so that each replicate with chemical inhibitors represents a completely independent experiment.

Unless otherwise indicated, chemical perturbations were done using the following chemical concentrations: 1 µM oligomycin, 1 µM FCCP, 1 µM TPB (Sigma-Aldrich), 5 µM EIPA (Sigma-Aldrich), 1 µg/ml nocodazole (Sigma-Aldrich), 5 µM STLC (Sigma-Aldrich), 15 µM proTAME, 1 µM RO-3306, 400 nM BMS-265246, 100 nM okadaic acid, 2 µM rotenone, 2 µM antimycin A, and 2 mM thymidine. All chemicals were diluted in DMSO, except okadaic acid, which was diluted in EtOH. Unless otherwise indicated, all chemical inhibitors were obtained from Cayman Chemicals. For cell cycle targeting chemicals, treatment efficiency was validated by examining the cell cycle profile.

**Validating CDK1 activity levels.** CDK1 activity was measured using two approaches, western blotting and flow cytometry, both based on labeling proteins using MPM2 antibody (monoclonal mouse antibody, EMD Millipore, #05-368). L1210 cells were first treated with chemicals indicated in Fig. 2b for 30 min, after which cells were washed with PBS and immediately lysed in XT sample buffer (Bio-Rad). Samples were sonicated to breakdown DNA, mixed with XT reducing agent (Bio-Rad), and loaded on a 4–12% Bis-Tris Criterion XT precast gel (Bio-Rad). Proteins were separated by running the gels in MES buffer (Rio-Rad) for 45 min at 180 V. The proteins were then transferred on a nitrocellulose membrane by blotting for 2 h at 40 V in Tris/Glycine buffer (Bio-Rad). The membrane was blocked by incubating with 5% BSA in TBST for 30 min in RT. CDK1 target protein phosphorylation was detected using MPM2 antibody (used at 1 µg/ml) and β-tubulin III was used as a loading control (polyclonal rabbit antibody, Sigma-Aldrich, #T2200) (used at 0.4 µg/ml). MPM2 and β-tubulin antibodies were labeled using fluorescent secondary antibodies (IRDye 680RD goat anti-rabbit secondary antibody, LI-COR, #925-68071; and IRDye 800CW donkey anti-mouse secondary antibody, LI-COR, #925-32212). The primary antibodies were incubated o/n in +4 °C, followed by three washes with TBST and a 1-h incubation with secondary antibody. Following TBST washes of the secondary antibody, the membrane was imaged using LI-COR Odyssey detection system. For flow cytometry-based quantifications, cells were treated with chemicals as for western blotting, washed with PBS, fixed in 4% PFA for 10 min and permeabilized with 0.5% Triton X-100 for 10 min. The cells were then washed with PBS and blocked with 5% BSA in PBS for 30 min. This was followed by MPM2 antibody labeling (used at 10 µg/ml) o/n in +4 °C. Following PBS washing, the MPM2 antibody was then labeled using Alexa Fluor 488-conjugated anti-mouse secondary antibody (Cell Signaling Technology, #4408) for 1 h at RT, after which the cells were washed three times with PBS. The MPM2 labeling was quantified using BD Biosciences flow cytometer LSR II HTS.

**Cell cycle synchronizations.** In L1210 and BaF3 cells, the cell cycle was synchronized to G2 using a double-thymidine block followed by G2 arrest. Briefly, cells were treated with 2 mM thymidine for 15 h, washed twice with PBS, cultured for 6 h in normal culture media, retreated with 2 mM thymidine for 6 h, washed twice with PBS, returned to normal culture conditions for 3 h and treated with 5 µM RO-3306 for 7 h. At this point, ~80–90% of cells were arrested in G2, and upon removal of RO-3306 part of the cells enter mitosis immediately[24].

The cell cycle status of L1210 cells was analyzed as follows: The cells were washed with PBS, fixed in 4% PFA for 10 min, washed with PBS, permeabilized with 0.5% Triton X-100 for 10 min, washed with PBS and blocked with 5% BSA in PBS for 30 min. The cells were then stained with p-Histone H3 (S10) antibody (D2C8, conjugated to Alexa 488, Cell Signaling Technology, #3465 S) in a PBS solution containing 5% BSA o/n at +4 °C. The p-Histone H3 antibody was used in the concentration recommended by the supplier. The following day the cells were washed with PBS and stained with 1:2000 dilution of NuclearMask Blue (#H10325, Thermo Fisher Scientific) for 30 min in RT. Finally, the cells were washed three times with PBS, mixed into PBS supplemented with 1% BSA, and put on ice until FACS analysis. The relative proportions of mitotic (4 N DNA content, p-Histone positive), G2 cells (4 N DNA content, p-Histone negative) and G1&S cell (<4 N DNA content, p-Histone negative) was quantified using BD Biosciences flow cytometer LSR II HTS with excitation lasers at 355, 488, and 561 nm, and emission filters at 450/50, 530/30, and 585/15. For quantification of cell cycle status in BaF3 and DT40 cells, we used propidium iodide (PI) staining of DNA in cells fixed with ice-cold 70% EtOH o/n. DNA content was quantified using BD Biosciences flow

cytometer LSR II HTS. See (Supplementary Fig. 9a, b) for representative FACS plots with gating.

**Oxygen-consumption measurements.** Oxygen consumption was measured using the XF24 Seahorse Extracellular Flux Analyzer. The Seahorse cell culture plates were coated with poly-L-lysine, and ~80,000 L1210 cells were loaded into each well. The experiment was started 3 h after plating the cells. The oxygen-consumption measurements were carried out in normal cell culture media (see above) to maximize comparability to the conditions within SMR. Oxygen consumption was measured four times with 6-min intervals, after which cells were treated with 1 µM oligomycin and, four measurements later, with 2 µM rotenone and 2 µM antimycin A. After the experiment, cells in each well were counted, and this cell count was used to normalize the data.

When measuring oxygen consumption in cells arrested in G2 and mitosis, G2-arrested cells were washed and moved back into 5 µM RO-3306 containing media (to maintain G2 arrest) or into 5 µM STLC containing media (to achieve prometaphase arrest)[24]. The cells were then plated on to the Seahorse cell culture plates, and 2 h later the oxygen-consumption measurement was carried out as described above. During the oxygen-consumption measurements, parallel cultures were processed for FACS-based cell cycle analysis as described above. The relative proportions of G2 and mitotic cells in the populations were used to normalize the oxygen-consumption data to account for incomplete cell cycle synchrony.

**Lactate efflux rate measurements.** Lactate efflux rate was measured from cells arrested in G2 and from cells released in free mitosis by quantifying lactate amounts in media using CG-MS. Cells were first synchronized to G2 using a double-thymidine block followed by a RO-3306-mediated G2 arrest, as described above. To release cells from G2 arrest, cells were spun down 500g at 37 °C for 5 minutes and washed once in media lacking RO-3306. Cells were spun down once more at 600 × g for 1 min and resuspended in RPMI with 11 mM glucose, 2 mM glutamine, 1 mM U-$^{13}$C-pyruvate, 10% dialyzed FBS, and penicillin/streptomycin with 10$^6$ cells/ml. One flask of cells of each biological replicate (separate synchronization) was immediately spiked with 5 µM RO-3306 to maintain G2 arrest. Media samples were collected at 19, 32, and 45 minutes after release from G2 arrest by spinning the sample at 500 × g for 3 minutes at 4 °C to pellet cells. The supernatants were transferred to a new tube and stored at −20 °C. Parallel cell cultures were used for cell counting and for FACS-based cell cycle analysis as described above.

For GC-MS, 150 µl of cell-free media was extracted in 800 µl of methanol containing 1.25 µM $^{13}$C1-sodium lactate to serve as an internal standard. Samples were spun down at 15,000 × g for 10 min at 4 °C and 400 µl of the supernatant was transferred to a new tube for derivatization. Samples were dried down under nitrogen gas. Dried samples were derivatized with 16 µl of methoxamine (MOX) reagent (Thermo Fisher, #TS-45950) and 20 µl of N-tert-butyldimethylsilyl-N-methyltrifluoroacetamide with 1% tert-butyldimethylchlorosilane (Sigma-Aldrich, #375934). Following derivatization, samples were analyzed using a DB-35MS column (30 m × 0.25 mm i.d. × 0.25 µm, Agilent J&W Scientific) in an Agilent 7890 gas chromatograph (GC) coupled to an Agilent 5975C mass spectrometer (MS). Data were analyzed and corrected for natural isotope abundance using in-house algorithms. The lactate efflux rate was obtained by a linear fit across the three timepoints for each independent replicate. Cell counts and the relative proportions of G2 and mitotic cells in the populations were used to normalize the data to account for incomplete cell cycle synchrony.

**ATP- and ADP-level measurements.** For ATP- and ADP-level measurements, cells were synchronized and processed identically to that used for lactate efflux rate analysis above. Thirty minutes after releasing the cells from G2 arrest, 100 µl of media containing 10$^6$ cells/ml were collected and flash-frozen. Intracellular ATP and ADP measurements were collected on 10 µl of thawed media in technical duplicates using an enzymatic, Luciferase-based assay according to the manufacturer's instructions (Sigma-Aldrich, #MAK135). Measurement linearity was established using an external ATP standard. Cell counts and the relative proportions of G2 and mitotic cells in the populations were used to normalize the data to account for incomplete cell cycle synchrony.

**Data analysis.** All details of the electrical circuit model and data analysis for the model can be found in (Supplementary Notes 3–8). The modeling and SMR & fluorescence data processing were carried out using custom MATLAB codes. Quantifications of TMRE increase during mitosis was carried out by normalizing the TMRE signal of each data point to the median signal a 3 h time period prior to daughter cell abscission, after which the highest TMRE value during mitosis was used for quantification.

**Statistics.** In population-level experiments, such as cell cycle analyses or oxygen-consumption measurements, each replicate represents an independent culture. In control SMR experiments, cell progenies were monitored for various durations, so that each independent experiment can have one or more mitotic events. By contrast, all chemical treatments within the SMR were carried out so that only one mitotic event was analyzed for each experiment. Details of replicate numbers can be found in

figures and figure legends. The experiments used for modeling were repeated over 30 times. Importantly, as some data were excluded from the modeling, as cells did not reach a steady TMRE baseline prior to cell division. For full details of data exclusion, please see (Supplementary Note 5). Statistical significances are indicated in each figure and the statistical details can be found in the figure legends. For statistical analysis of the electrical circuit model results, please see (Supplementary Note 8). All statistical analyzes were carried out using MATLAB or Origin Pro 2019.

**Reporting summary**. Further information on research design is available in the Nature Research Reporting Summary linked to this article.

## Data availability
The authors declare that all data supporting the findings of this study are available in the paper or in its Supplementary section. Data for single-cell TMRE traces are included in the Electrical circuit analysis code file. Source data are provided with this paper.

## Code availability
The authors declare that all new analysis codes required for reproducing this paper are available in the Supplementary section. MATLAB code used to analyze single-cell TMRE traces is included in the Supplementary section in the Electrical circuit analysis code file.

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

## Acknowledgements

We would like to thank Douaa Mugahid for helpful comments. This research was supported by a Samsung Scholarship (J.H.K.), the Koch Institute Frontier Research Program through the Kathy and Curt Marble Cancer Research Fund (S.R.M.), the Cancer Systems Biology Consortium funding (U54-CA217377) from the National Cancer Institute (S.R.M.), the MIT Center for Precision Cancer Medicine and National Institute of Health grants P30-CA14051, R01-GM104047 (M.B.Y.), R35-ES028374 (M.B.Y.), R35-CA242379 (M.G.V.H.), R01-CA201276 (M.G.V.H.), and by the Wellcome Trust grant 110275/Z/15/Z (T.P.M.). M.G.V.H. also acknowledges support from the Lustgarten Foundation, the Ludwig Center at MIT, and a faculty scholars award from HHMI.

## Author contributions

J.H.K., M.A.S., and T.P.M. planned and carried out the SMR experiments. G.K. carried out the modeling and image analysis. Z.L. and K.M.S. carried out the metabolite-level measurements. T.P.M. carried out the imaging and oxygen-consumption measurements. D.L. and T.P.M. generated the reporter cell lines. J.H.K., G.K., and T.P.M. analyzed the data. J.H.K., G.K., and T.P.M. wrote the paper with input from all authors. M.G.V.H., M.B.Y., S.R.M., and T.P.M. supervised the work and provided funding for the experiments. T.P.M. conceived the study.

## Competing interests

S.R.M. is a co-founder of Travera and Affinity Biosensors, which develops techniques relevant to the research presented. M.G.V.H. is on the scientific advisory board of Agios Pharmaceuticals, Aeglea Biotherapeutics, iTEOS therapeutics, and Auron Therapeutics. The remaining authors declare no competing interests.
