## [Peer Review File · Nature Communications]

Reviewers' comments:

Reviewer #1 (Remarks to the Author):

Kang and colleagues present here new findings proving that the inhibition of mitochondrial ATP synthesis is dispensable for mitosis in leukemia cells. Most of the results are based on a method previously developed by the authors (microfluidic mass sensor) that when coupled to a fluorescence detection system allows for the quantification at the single cell level of the mitochondrial membrane potential. Moreover, they developed an electrical circuit model to predict the activity of the ATP synthase. This is an original manuscript that has some novelty in the methods that are used. In the biological relevance, however, the novelty is precluded by an extensive number of studies that show similar data. The results are not clearly explained, and in general the paper is difficult to follow. What each individual experiment indicates and why (the rationale) is not explained. There are plenty of examples all over the manuscript. For instance, for non-cell cycle experts, it should be explained that Geminin is degraded during the metaphase to anaphase transition. Indeed, the fourteen supplementary figures and the ten supplementary notes do not help to improve the complexity of the manuscript, which under the opinion of this reviewer is too high for a journal of general interest. Moreover, there is extensive literature addressing all the changes of mitochondrial function and dynamics that take place during the cell cycle progression, which merit, at least to be discussed and cited both in the Introduction and in the discussion sections (the discussion cannot be limited to two paragraphs). In particular, I have the following concerns:

1. The authors claim that their results challenge the concept that mitosis requires high energy levels (ATP). They show that ATP synthase is inhibited during mitosis. Anaerobic glycolysis could compensate for the decreased mitochondrial ATP synthesis. Incubation of the cells without glucose and in the presence of galactose would provide information about this alternative to ATP synthesis. If ATP is not required, then cells could still terminate the cell cycle in the absence of glucose, when incubated with galactose and oligomycin. Otherwise, oligomycin would arrest the cells in mitosis.
2. Despite the fact that the technique used to predict and model the rate of ATP synthesis is novel, it should be validated by the measurement of the actual ATP levels by biochemical techniques.
3. It is not clear how the authors make the correlation of mitochondrial membrane potential and activity with the distinct cell cycle phases. Is this performed by FACS analyses? And if yes, where is this data shown?
4. In Figure 3, the results have to be confirmed using alternative methods, such as FACS analyses or cell counting (3A). An extended time should be tested. Moreover, cell death could mask the results of this experiment. Have the authors quantified cell viability in response to the treatments?
5. In Figure 1, the authors show that mitochondria hyperpolarize during prophase and metaphase, but they do not provide any biological evidence that the cells that they are analyzing are indeed in these specific mitotic phases. Since some of their conclusions are based on these experiments, it is important that more evidence is provided (imaging, markers of the mitotic phases, ...).
6. The authors show that CDK1 activity drives the hyperpolarization of mitochondria in early mitosis. But this has been extensively published. Indeed, it is well established that CDK1 phosphorylates and regulates the activity of some components of the ETC in the mitochondria, thus promoting oxidative phosphorylation and ATP synthesis during mitosis (Wang et al. (2014) Cyclin B1/Cdk1 coordinates mitochondrial respiration for Cell-cycle G2/M progression. *Dev. Cell* 29, 217–232). It is even explained in the textbooks (Alberts, B. et al. *The cell-cycle control system in Molecular Biology of THE CELL* (Fifth Edition). 1060–67).

7. In Figures 2B-C the inhibition of CDK1 should be proved. The authors say that CDK1 is partially inhibited, but they do not show it.

Reviewer #2 (Remarks to the Author):

This paper reports an impressive high-tech piece of quantitative bioenergetics at the single cell level. The experimental design and data quality look very solid and exciting, especially the suspended microchannel resonators (SMR) used as non-invasive single-cell buoyant mass sensor and its combination with fluorescence detection. Also, important controls were done to validate techniques, and an attempt is made to model the system to extract quantitative bioenergetic parameters from their data.

Importantly, the authors are aware of, and honest about, the limitations of the techniques for both experiments and the electrical circuit model. I am sure there are steps in the analysis that can be criticized, such as the choice for a biphasic, discrete model to estimate time scales, or the exact error propagation model, but the choices are clear and well documented - although not always clearly justified, see below. The other main issue with the paper is the lack of proper discussion and embedding in the current literature, it is now a very technical paper.

The discussion itself is really short, and strangely naïve and speculative after such a solid and elaborate effort to get the parameters. If there is 4 mM of ATP, one can relatively easily compare this to the bioenergetic demands of cells, overall, and from there estimate the demand during mitosis. Turnover of ATP is a matter of seconds to minutes, and so a complete block of ATP synthesis would be a disaster, I would think. I therefore suspect that the cells rely on aerobic glycolysis. One could measure lactate production or enhanced glucose consumption to test this. The authors should at least substantiate their speculations with proper calculations, or check other studies that used oligomycin.

ATP is also not the only factor that affects cellular bioenergetics, also ADP, and AMP (energy charge) do: many enzymes have micromolar affinities for ADP as well. Overall I found the discussion about the bioenergetic consequences of the ATP synthesis inhibition to be very weak. The authors should do a better job in selling why their data are so relevant, new or surprising, biologically.

With respect to the model, I miss a discussion on described effects of oligomycin on cell physiology, to substantiate the essential assumptions that form the basis for the ATP production rate calculations. The model is clear, but justification of the model simplifications can be improved, as it seems based on papers from one group (Nicholls). It is not my specific field, but I wonder how widely accepted the circuit model really is. For example, I miss thermodynamic or other physiological justifications of the irreversibility of the ETC. Surely many bacteria rely on reversal of this process to generate reducing power in the form of NAD(P)H during photosynthesis (called reversed electron flow)!

I also miss the biological interpretation of capacitances, or an argument why it would be constant. There is actually a lot of "which we assumed" in the model. Ideally, any time you read this, you either expect a good argument/justification/reference, or a sensitivity analysis to show it does not affect the conclusions.

Reviewer #3 (Remarks to the Author):

The authors show that in murine lymphocytic leukemia cell line L1210, as well as other murine cell lines, the mitochondrial membrane potential oscillates. More specifically, mitochondrial

hyperpolarization occurs shortly after the onset of prophase, peaks at the transition from meta- to anaphase and ends before cytokinesis, as was determined relative to mitotic events with known timing. Furthermore, by use of chemical inhibitors, it is determined that the mitochondrial hyperpolarization and its spike-like behavior depend on CDK1-activity. The authors also use chemical inhibitors to show that mitochondrial ATP synthesis is required for cell growth, but not for mitotic entry and progression of mitosis. Finally, ATP synthesis rates during mitosis were derived through modelling and verified using oxygen consumption rate measurements. These efforts indicate that ATP synthesis rates drop during mitosis.

The paper draws on two highly timely tools: (i) highly sophisticated tools for single cell analyses, and (ii) integration of dynamic data into a mathematical model. While biology is full of state measurements, here the authors quantified something that is much harder to determine, namely a rate (i.e. the one of ATP synthesis), which they do even in a manner that is time resolved. This is by far not trivial and is a major achievement by itself. Overall, I feel that this work could become a contribution for Nature Communications, if the following concerns can be addressed:

Major concerns:

1. Is there a crucial control missing? The authors draw several conclusions by the use of inhibitors (cf. Fig. 2). These inhibitors were dissolved in DMSO or EtOH (line 373-374). Could it be that the observed effect is only due to the vehicle? Isn't it necessary to show vehicle controls?

2. Strange finding: expression of metabolic reporters completely alters behavior: The authors state that expression of exogenous metabolic FRET-based reporters alters mitotic behavior of the mitochondrial membrane potential. According to the authors, the strong CMV promoter used to express the reporters could cause the change in mitochondrial metabolism.

a) Why do the authors not use a weaker promoter to express these reporters? Why does such a strong expression of an exogeneous protein lead to a completely different behavior? What kind of conclusions can be drawn from this for the system that they are studying?

b) More crucially, however, N-terminal tagging of geminin with monomeric Azami-Green (mAG) under control of the endogenous promoter also brings about a change in the mitochondrial hyperpolarization during mitosis (fig. S9B/C). Thus, it is unclear whether the changes in metabolism are caused by highly expressed exogenous proteins or by expression of fluorescent proteins in general.

3. Are model conclusions robust? The model is a central element on the basis of which they obtained the final conclusion of their paper namely that mitochondrial ATP synthesis rates drop during mitosis.

a. The authors assumed a binary state in their CDK activity (either on or off). Also, for their analyses, they assumed times when the CDK1 activity would either be on or off. It is not clear in how far their results would change when these times are changed. Or when the CDK1 activity would be considered to have more gradual dynamics. Would the outcome/conclusions change?

b. Fig. 4F: it is not clear to me how the authors determine the ATP synthesis in G2 from the single-cell data, as they fit their model only to the transient $\Delta\psi$ dynamics, which is not in G2.

c. Can one really compare the single-cell and the population-level experiments as shown in Fig. 4F?

d. Can the modelling results shown in fig. 4G be verified experimentally, e.g. with another inhibitor that leads to an anaphase block (potentially proTAME)?

4. Dynamics of ATP levels is a crucial, but lacking piece of information in the work: A key element that is missing in the story are data on ATP levels before, during and after mitosis. The reason for this is the following: On the one hand, the authors state in the discussion that the turnover rate of the ATP is in the minute range, meaning that if the synthesis of ATP would be stopped, ATP would drop to zero in a few minutes. In the discussion, the authors state that in mitosis there is significant ATP consumption (i.e. they stick to the current notion that mitosis requires lots of ATP). In their work, they claim that ATP synthesis drops in mitosis. Taking these 3 pieces of information together, this would mean that ATP levels NEED to drop during mitosis, if they are right with their finding / if the current notions is

correct. If a measurement of ATP levels during mitosis would not show any drop, this would mean that either their finding is not correct, or that the current notion (i.e. mitosis requiring lots of ATP) is incorrect. In any case, this would advance our understanding tremendously. Notably, while many papers working on energy homeostasis measure metabolite level dynamics, and then to guess-work on the rates of the underlying processes, here the authors did the much harder thing (i.e. they quantified a rate), but did not look at the dynamics of the state variable (i.e. ATP) itself. I feel that the authors should measure in synchronized cultures (potentially by means of metabolomics) the levels of ATP during the cell cycle. This would make the "package" complete.

Minor concerns:

1. I find the discussion section weak and not adequate for a high profile paper: What are the main advances? What are the key aspects that will require further investigations? What are the key implications of the work?
2. For some key data, only data from an individual cell is shown, e.g. Fig. 1B, C and D. Is it possible to show data from more cells (or more cell cycles); ideally like the cell trace shown in Fig. 1A. These are key figures and I would not feel comfortable only having seen data from an individual cell from an individual cycle.
3. Comparing fig. 2B and E, I notice a difference between the 2 STLC+RO-3306 conditions in TMRE/mass signal at equilibrium. When STLC and RO-3306 are added simultaneously, the TMRE/mass signal at equilibrium is higher than during G2, whereas if RO-3306 is added only after the STLC treatment equilibrium has been achieved, the TMRE/mass signal drops back to (or below) the G2 state. How can this difference be explained?
4. Page 9, line 170ff: This experiment assumes that the arrest does not affect the metabolic rates. Can this assumption be safely made? What if it is not true?
5. Goodness of fit (R^2 ; model to experimental data) varies for each single cell trace (fig. S11A). Is there a general indication of R^2 to be given?

Suggestions:

1. fig. 2B/C: change order
2. fig. 3A: show typical no treatment cell traces to make figure interpretation more intuitive.
3. fig. 3C/D: extend the x-axis to $t = -15$ min to show start of oligomycin treatment.
4. fig. 3E: change the y-axis label of the boxplot to Growth rate $t = 1$ h .
5. Page 7: lines 110f: Mention that the oxygen consumption measurements were done on synchronized cells. This will help the reader.
6. Fig. 4G: I find "ATP synthesis amount" (also somewhere mentioned in the text) a strange term
7. Fig. 4F: what are the n 's indicated for the OCR experiments? Are these independent synchronizations and measurements?
8. Page 9, line 175: leakage of what?
9. Page 5, line 100: "we observed that PARTIAL CDK1 inhibition..."
10. In figure 1, "non-quenching" and "quenching" concentrations are highlighted. For a reader not particularly familiar with the used dyes, this might be difficult to grasp. Thus, to enhance accessibility for a broader readership, I suggest that the author explain this, also with a short comment in the main text.
11. The authors find that certain mouse lymphocytes do not show the characteristic spike in $\Delta\psi_m$ increase. Could they speculate why this is the case, and whether this provides any further insight for their story? Would these cells have hardly any respiratory activity to start with?

Corrections:

1. Line 137: analyses (instead of analyzes)
2. Line 395: G1/S cells are described as having $>4N$ DNA content, this should be $<4N$.
3. Page 5, line 96: Reference to figure missing
4. Fig. 2B: vertical dashed line missing
5. Fig. 2B: legend in the figure and the caption text do not match with regards to the colors mentioned

6. Page 11, line 199: Reference needed for the Km values being in the micromolar range. (I am in fact not certain whether this statement (without reference) is really correct).
7. Fig. 3C/D: Specify what the control cells were

Reviewer #4 (Remarks to the Author):

The authors seek to quantify two mitochondrial parameters during mitosis: 1) membrane potential and 2) rate of mito ATP synthesis.

For #1, they apply a neat method to measure mass of cells to then compute mass-normalized mito membrane potential at high temporal resolution in asynchronous, normally dividing single cells. They report acute mitochondrial hyperpolarization during mitosis, with a peak at the metaphase-anaphase transition. These data need further support and refined interpretation:

- 1) The current experiments rely on whole-cell TMRM fluorescence. The authors should sort the cell populations of interest and measure membrane potential in permeabilized cells and in whole cells by microscopy for finer spatial resolution. Although permeabilization is artificial, it may help reinforce their claims and overcome confounders such as mitotic swelling and plasma membrane hyperpolarization.
- 2) The authors seem to be wholly unaware of the landmark work of Toren Finkel and Lippincott-Schwartz — both prominent in this field — that have written key papers within the past decade on the coordination of mitochondrial bioenergetics with the cell cycle. Are the authors unaware of this work which is foundational for the current paper?!
- 3) Informed with this literature, the authors must perform microscopy analysis — using TMRM staining and mitochondrial markers as a function of time — to determine whether the membrane potential signal is simply an artifact of mitochondrial fusion and fission.
- 4) There are two cases where the authors do not observe said mitotic mitochondrial hyperpolarization i) mouse FL5.12 lymphocytes and ii) cells over-expressing mitochondrial reporter constructs. The authors should try to explain these notable exceptions.

For #2, the authors model the mitochondrion as a circuit and derive the rate of mitochondrial ATP synthesis. Their analytical and empirical data from the circuit-based approach is supported by supplementary traditional OCR measurements. Altogether, the authors report reduced mitochondrial ATP synthesis. However, extending this observation to “challenge the traditional dogma that cell division is a highly energy demanding process.” (abstract) and “suggests a much lower rate of ATP consumption during mitosis than previously assumed” (discussion) is premature as it wholly ignores ATP demand and it wholly ignores glycolytic ATP synthesis. For clarity -- if the cell is not using ATP (no load on the resistor), there is no need for mito ATP production. For clarity -- if the cell is doing plenty of glycolysis, there will be plenty of glycolytic ATP and the cells will not need to do mito ATP OXPHOS. Recall respiratory control, mito respiration is primarily regulated by ADP availability.

Reviewer response letter by Kang, Katsikis, et al.

Summary

We would like to thank the editor and the reviewers for their constructive feedback. We have now addressed the reviewer comments with many new experiments and also by explaining our work better through improved writing. Our new experiments have validated our previous findings as well as expanded the scope of our work. In our revised manuscript, all relevant changes are marked with grey highlighting. Please see below for our list of most major changes as well as our point-by-point responses to reviewer's comments. All our responses are in blue font, while reviewer comments are in black.

Key changes:

1. Glycolysis rates and ATP & ADP levels during mitosis: We have now measured ATP and ADP levels as well as lactate efflux rates from cell synchronized to G2 and mitosis. These experiments revealed a decreased lactate efflux rate and decreased ATP & ADP levels in mitosis (Figure 5). We discuss these results extensively in our revised discussion. Overall, these new data validate our original conclusion about reduced ATP synthesis during mitosis, and they also allow us to conclude that ATP consumption is lower in mitosis than in G2.
2. Validation & control experiments for CDK1 inhibition: As requested by reviewers #1 and #3, we have now validated how our chemical perturbations influence CDK1 activity, and we have also added control data to our experiments where we perturb CDK1 activity mid-mitosis.
3. Membrane potential and mitochondria morphology during cell cycle: As requested by reviewer #4, we have now carried out imaging experiments of L1210 cell mitochondrial morphology in mitosis (Figures 1H and S7). This revealed that the L1210 cell mitochondria do not undergo extensive fission in mitosis. In addition, we have validated that the TMRE signal is dominantly coming from mitochondria. We also now acknowledge that mitochondrial morphology can influence mitochondrial activity, but this does not influence our conclusions about ATP synthesis rate changes.
4. Rewriting of the manuscript: We have now extended our introduction and discussion sections to better embed our work in the current literature. We have also improved the clarity of our results section for non-expert readers.
5. Title: As several reviewers criticized our work for being too technical, and as the ATP level and glycolysis rate measurements expand the scope of our work, we have changed our title accordingly to "Monitoring and modeling of bioenergetics reveals decreased ATP synthesis during cell division".

Point-by-point responses to reviewer comments

Reviewer #1 (Remarks to the Author):

Kang and colleagues present here new findings proving that the inhibition of mitochondrial ATP synthesis is dispensable for mitosis in leukemia cells. Most of the results are based on a method previously developed by the authors (microfluidic mass sensor) that when coupled to a fluorescence detection system allows for the quantification at the single cell level of the mitochondrial membrane potential. Moreover, they developed an electrical circuit model to predict the activity of the ATP synthase. This is an original manuscript that has some novelty in the methods that are used. In the biological relevance, however, the novelty is precluded by an extensive number of studies that show

similar data. The results are not clearly explained, and in general the paper is difficult to follow. What each individual experiment indicates and why (the rationale) is not explained. There are plenty of examples all over the manuscript. For instance, for non-cell cycle experts, it should be explained that Geminin is degraded during the metaphase to anaphase transition. Indeed, the fourteen supplementary figures and the ten supplementary notes do not help to improve the complexity of the manuscript, which under the opinion of this reviewer is too high for a journal of general interest. Moreover, there is extensive literature addressing all the changes of mitochondrial function and dynamics that take place during the cell cycle progression, which merit, at least to be discussed and cited both in the Introduction and in the discussion sections (the discussion cannot be limited to two paragraphs). In particular, I have the following concerns:

We would like to thank the reviewer for the constructive and useful feedback. We apologize for the lack of clarity in our manuscript, which we have now improved significantly. We have now also extended our introduction and discussion sections to cover other areas of mitochondrial biology than just ATP synthesis. These new areas we discuss include mitochondrial dynamics, as suggested by the reviewer. Yet, we disagree with comment “novelty is precluded by an extensive number of studies that show similar data”, and we discuss this in detail under the reviewer’s comment #6. Overall, thanks to reviewer’s comments, we feel the quality of our manuscript has significantly improved and we are grateful for the constructive feedback.

1. The authors claim that their results challenge the concept that mitosis requires high energy levels (ATP). They show that ATP synthase is inhibited during mitosis. Anaerobic glycolysis could compensate for the decreased mitochondrial ATP synthesis. Incubation of the cells without glucose and in the presence of galactose would provide information about this alternative to ATP synthesis. If ATP is not required, then cells could still terminate the cell cycle in the absence of glucose, when incubated with galactose and oligomycin. Otherwise, oligomycin would arrest the cells in mitosis.

We thank the reviewer for suggesting this. We fully agree that our original manuscript did not address if glycolytic ATP synthesis could compensate for the decreased mitochondrial ATP synthesis in mitosis. We have decided to more directly investigate glycolytic rates by comparing lactate efflux in cells in G2 and in mitosis. This was also suggested by reviewers #3 and #4. Our results show that mitotic cells have ~50% lower lactate efflux rate compared to G2, suggesting decreased glycolytic ATP synthesis rates in mitosis. This, together with our measurements of mitochondrial ATP synthesis rates, argues that the overall ATP synthesis is lower in mitosis than in G2. As cellular ATP pools are fully consumed in ~1 min in interphase, the overall decrease in mitotic ATP synthesis can only take place if ATP consumption also decreases in mitosis. Thus, these results support our original conclusion that cell division is not highly energy consuming in comparison to interphase. Please see our results section (starting on line 293), Figure 5C and our discussion (starting on line 329) for full details.

In addition, we have also carried out an experiment similar to that suggested by the reviewer. We synchronized L1210 cells to G2 and released the cells into mitosis in the presence or absence of glucose. This revealed that glucose was not required for mitotic entry, but it was necessary for mitotic progression/exit. However, we would like to point out that all experiments aiming to perturb glycolytic rates lack specificity towards glycolytic ATP synthesis independently of other glycolytic products. Thus, this experiment only proves that glycolysis as a whole is required for mitotic exit, and we present these results in our supplement. Please see our results section (starting on line 218) and Figures S9G,H.

2. Despite the fact that the technique used to predict and model the rate of ATP synthesis is novel, it should be validated by the measurement of the actual ATP levels by biochemical techniques.

We fully agree with the reviewer and we have now measured ATP levels using a standard luciferase-based biochemical assay in cells synchronized to G2 and mitosis. This revealed that ATP levels

decrease ~40% in mitosis and ADP levels decrease approximately equally. As we already pointed out in our manuscript, decreased ATP levels in mitosis have also been observed in previous literature. Please see our results section (starting on line 297) and Figures 5D,E.

3. It is not clear how the authors make the correlation of mitochondrial membrane potential and activity with the distinct cell cycle phases. Is this performed by FACS analyses? And if yes, where is this data shown?

We apologize for the lack of clarity on this. We have now improved the clarity of results section so that the reader doesn't have to rely on supplemental information & methods section. In short, as we monitor individual cells progressing through the cell cycle, we have simultaneously monitored fluorescent (FUCCI cell cycle marker) and biophysical (cell density and stiffness) cell cycle markers, all of which have previously been shown to change in specific mitotic stages (i.e. these are markers of specific mitotic stages). Our results section now clarifies this at several points (starting on line 118). Furthermore, our claims about the $\Delta\Psi_m$ increasing during specific stages of mitosis are further supported by our cell cycle arrest in G2 (Fig. 1D, RO3306 treatment), in prometaphase (Fig. 2A, STLC and nocodazole treatments) and in metaphase (Fig. 2A, proTAME treatment).

In addition, we have now carried out a more classical comparison between $\Delta\Psi_m$ and cell cycle stage, by comparing how MitoTracker Red CMXRos ($\Delta\Psi_m$ sensitive probe) signal differs between mitotic and G2 cells. Mitotic and G2 cells were separated using DNA content, Histone H3 (Ser10) phosphorylation and Cyclin-B levels. Using this approach we observed that mitotic cells have higher levels of $\Delta\Psi_m$ (text starting on line 133 and Figures S5E,F), which is fully consistent with our previous observations.

4. In Figure 3, the results have to be confirmed using alternative methods, such as FACS analyses or cell counting (3A). An extended time should be tested. Moreover, cell death could mask the results of this experiment. Have the authors quantified cell viability in response to the treatments?

We agree with the reviewer that results presented in our original Figure 3A do not alone prove that cells divide normally in the presence of oligomycin. To further support this, we carried out additional experiments using FACS-based cell cycle analysis (displayed in original Figure 3C and 3D) to validate that mitosis is not inhibited by oligomycin.

Following the reviewer's suggestion, we have now also tested if mitotic progression is influenced by extended oligomycin treatment time before releasing cells in to mitosis. The results (Figure R1, below) still indicate that oligomycin treatment has little impact on cell division. However, as it is well-known that oligomycin stops oxygen consumption in minute time scales (Figure S8A), we do not see the necessity of including extended oligomycin treatment results in our manuscript to validate that mitochondrial ATP synthesis is not essential for mitosis. However, we now specify in our results section (starting on line 213) that "mitochondrial ATP synthesis is not acutely required to support cell division" to account for the possibility that more prolonged oligomycin treatments could have different results.

Regarding cell viability, we have now quantified if cell cycle synchronization or oligomycin treatment result in increased cell death by examining the prevalence of cells with sub-G1 DNA content (i.e. apoptotic particles). We did not observe any statistically significant difference between control and oligomycin treated cells and overall fraction of cells with sub-G1 DNA content remained low (Figure R2, below). We now include example FACS plots displaying DNA content at different timepoints during the cell cycle synchrony experiments (Figures S9A,B).

Figure R1. L1210 cells were synchronized to G2 and released in to mitosis. Samples were collected at 30 min after G2 release for analysis of mitotic entry (left plot, displaying % of cells that have entered mitosis) and at 2.5h after G2 release for analysis of mitotic exit (right plot, displaying % of cells that have entered G1). Oligomycin treatment was started either 15min prior to mitotic release (as in our manuscript) or 4h prior to mitotic release. Despite prolonged oligomycin treatment, mitotic entry and exit are minimally affected. N=4 independently synchronized cultures. Data depicts mean \pm S.D. p-values were calculated using ANOVA followed by Holm-Sidak posthoc test.

Figure R2. L1210 cells were synchronized to G2 and released in to mitosis. Samples were collected at 30 min and 2.5h after G2 release. Oligomycin treatment was started 15min prior to mitotic release (as in our manuscript). Cell viability was analyzed based on % of particles with sub-G1 DNA content. We did not observe statistically significant changes in cell viability following cell synchronization or oligomycin treatment. N=4 independently synchronized cultures. Data depicts mean \pm S.D. p-value was calculated using ANOVA.

5. In Figure 1, the authors show that mitochondria hyperpolarize during prophase and metaphase, but they do not provide any biological evidence that the cells that they are analyzing are indeed in these specific mitotic phases. Since some of their conclusions are based on these experiments, it is important that more evidence is provided (imaging, markers of the mitotic phases, ...).

Thank you for pointing this out. We have indeed used various mitotic markers and we apologize for not explaining this clearly enough. Please see above our response to your point #3 for full explanation of how we detect mitotic stages.

6. The authors show that CDK1 activity drives the hyperpolarization of mitochondria in early mitosis. But

this has been extensively published. Indeed, it is well established that CDK1 phosphorylates and regulates the activity of some components of the ETC in the mitochondria, thus promoting oxidative phosphorylation and ATP synthesis during mitosis (Wang et al. (2014) Cyclin B1/Cdk1 coordinates mitochondrial respiration for Cell-cycle G2/M progression. *Dev. Cell* 29, 217–232). It is even explained in the textbooks (Alberts, B. et al. The cell-cycle control system in *Molecular Biology of THE CELL* (Fifth Edition). 1060–67).

We are grateful to the reviewer for the critical feedback, but we disagree about the novelty of our work. Indeed, Wang et al. (2014) have carried out an elegant study where they show that CDK1 phosphorylates and activates ETC complex I. We acknowledged and cited this paper several times throughout the manuscript. However, neither that study nor any other studies (according to our knowledge) have looked at the dynamics of mitochondrial membrane potential or ATP synthesis rates in mitosis. (Wang et al. provide extensive evidence for ETC activation, but they do not characterize ATP synthase activity changes). Furthermore, Wang et al. have the opposite conclusion from our study, as they claim an increased mitochondrial ATP synthesis in mitosis, whereas we show that mitochondrial ATP synthesis decreases using two fully independent approaches. The contradicting results may be due to differences between cell types studied, in which case we provide novel insights about mitochondrial ATP synthesis regulation in lymphocytes. Accordingly, we have now acknowledged in our discussion that our results about ATP synthesis are limited to one cell line (starting on line 340).

We have also looked through the textbook cited by the reviewer, but we were unable to locate any discussion about how or why mitochondrial energy metabolism would change in mitosis. We strongly believe that our study presents a unique approach to provide quantitative evidence on mitochondrial bioenergetics in mitosis, providing novel findings that extend our understanding of mitotic bioenergetics. We have now improved our introduction and discussion sections to better account for the work done before, and more specifically acknowledge that others have examined ETC activity in mitosis (referring Wang et al. (2014) as well as other papers that have measured ETC activity in mitosis) but that ATP synthesis rates have not been examined.

7. In Figures 2B-C the inhibition of CDK1 should be proved. The authors say that CDK1 is partially inhibited, but they do not show it.

The reviewer is correct about the lack of validation data and we thank him/her for pointing this out. We have now repeated our perturbations to CDK1 activity and monitored phosphorylation levels of CDK1 targets using MPM2 antibody. Our results section now states (starting on line 180) that “We validated that RO-3306 and BMS-265246 inhibit CDK1 activity by using western blotting with MPM2 antibody (Figure 2B), which identifies CDK1/2-phosphorylated sites found on various proteins^{38,39}. We also quantified the MPM2 antibody staining using flow cytometry (Figures 2C).”

Reviewer #2 (Remarks to the Author):

This paper reports an impressive high-tech piece of quantitative bioenergetics at the single cell level. The experimental design and data quality look very solid and exciting, especially the suspended microchannel resonators (SMR) used as non-invasive single-cell buoyant mass sensor and its combination with fluorescence detection. Also, important controls were done to validate techniques, and an attempt is made to model the system to extract quantitative bioenergetic parameters from their data. Importantly, the authors are aware of, and honest about, the limitations of the techniques for both experiments and the electrical circuit model. I am sure there are steps in the analysis that can be criticized, such as the choice for a biphasic, discrete model to estimate time scales, or the exact error

propagation model, but the choices are clear and well documented - although not always clearly justified, see below. The other main issue with the paper is the lack of proper discussion and embedding in the current literature, it is now a very technical paper.

We would like to thank the reviewer for the insightful and constructive feedback. We apologize for the shortness of our discussion and for the lack of justifications for some of our modeling approaches, both of which we have now improved significantly. We acknowledge that our manuscript has a technical feel to it – this is partly because we found it essential to discuss details and reliability of our unique approach to bioenergetics. Thanks to the improvements suggested by reviewer(s), we feel our manuscript has now found a better balance between technical rigor and broad & clear conclusions.

The discussion itself is really short, and strangely naïve and speculative after such a solid and elaborate effort to get the parameters. If there is 4 mM of ATP, one can relatively easily compare this to the bioenergetic demands of cells, overall, and from there estimate the demand during mitosis. Turnover of ATP is a matter of seconds to minutes, and so a complete block of ATP synthesis would be a disaster, I would think. I therefore suspect that the cells rely on aerobic glycolysis. One could measure lactate production or enhanced glucose consumption to test this. The authors should at least substantiate their speculations with proper calculations, or check other studies that used oligomycin.

We fully agree with the reviewer and we have now investigated glycolytic rates by comparing lactate efflux in cells in G2 and in mitosis. Our results show that mitotic cells have ~50% lower lactate efflux rate, suggesting decreased glycolytic ATP synthesis rates in mitosis. This, together with our measurements of mitochondrial ATP synthesis rates, argues that the overall ATP synthesis is lower in mitosis than in G2. As the reviewer also mentioned, cellular ATP pools are fully consumed in ~1 min during interphase and thus the ATP consumption needs to be decreased following the decrease in ATP synthesis in mitosis. Altogether, these results support our original conclusion that cell division is not highly energy consuming in comparison to interphase. Please see our results section (starting on line 293), Figure 5C and our discussion (starting on line 329) for full details.

ATP is also not the only factor that affects cellular bioenergetics, also ADP, and AMP (energy charge) do: many enzymes have micromolar affinities for ADP as well. Overall, I found the discussion about the bioenergetic consequences of the ATP synthesis inhibition to be very weak. The authors should do a better job in selling why their data are so relevant, new or surprising, biologically.

We fully agree with the reviewer and we have now measured ATP and ADP levels using a standard luciferase-based biochemical assay in cells synchronized to G2 and mitosis. This revealed that ATP levels decrease ~40% in mitosis and ADP levels decrease approximately equally. As we already pointed out in our manuscript, decreased ATP levels in mitosis have also been observed in previous literature. We now discuss these findings as well as their implications (impact on, for example, AMPK activation, allosteric regulation of enzymes and phase separation of disordered proteins) in our updated discussion. Please see our results section (starting on line 297), Figures 5D,E and our discussion (starting on line 357) for full details.

With respect to the model, I miss a discussion on described effects of oligomycin on cell physiology, to substantiate the essential assumptions that form the basis for the ATP production rate calculations.

We thank the reviewer for this helpful suggestion. We have now added more discussion about the physiological role of oligomycin to our supplemental section. In short, we acknowledge that cell's metabolism will be affected by oligomycin treatment, but our approach is designed to minimize bias caused by this. Please see our supplemental section (starting on line 250) for full discussion.

The model is clear, but justification of the model simplifications can be improved, as it seems based on papers from one group (Nicholls). It is not my specific field, but I wonder how widely accepted the circuit model really is.

The reviewer is correct that we rely heavily on papers from one lab (Nicholls). This is because the concept of mitochondria as an electrical circuit has been spearheaded by Prof. David Nicholls and it is discussed in his publications (several of which we cite) as well as in text books (e.g. Bioenergetics, 3rd Edition, David Nicholls and Stuart Ferguson). However, similar electrical circuit models/views have also been used by other labs working on mitochondrial bioenergetics¹⁻⁴, and mitochondrial specialists, such as Prof. Martin Brand (MRC Mitochondrial Biology Unit, Cambridge, UK), discuss mitochondria in the following terms: “By analogy to a simple electrical circuit, the inner membrane can be thought of as a non-ohmic proton conductor: the oligomycin-insensitive respiration used to drive proton leak (current) increases nonlinearly with respect to Δp (electrical potential)”⁵. We have now added several of the references above to our manuscript in order to acknowledge that the electrical circuit view of mitochondria is consistent with general view of mitochondrial bioenergetics. We now also discuss mitochondria in terms of an electrical circuit already in our introduction in order to make our modeling approach more easily accessible to broader audiences.

References:

- 1 Lemeshko, V. V. VDAC electronics: 3. VDAC-Creatine kinase-dependent generation of the outer membrane potential in respiring mitochondria. *Biochim Biophys Acta* 1858, 1411-1418, doi:10.1016/j.bbame.2016.04.005 (2016).
- 2 Padmaraj, D., Pande, R., Miller, J. H., Jr., Wosik, J. & Zagozdzon-Wosik, W. Mitochondrial membrane studies using impedance spectroscopy with parallel pH monitoring. *PLoS One* 9, e101793, doi:10.1371/journal.pone.0101793 (2014).
- 3 Diogo, C. V., Grattagliano, I., Oliveira, P. J., Bonfrate, L. & Portincasa, P. Re-wiring the circuit: mitochondria as a pharmacological target in liver disease. *Curr Med Chem* 18, 5448-5465, doi:10.2174/092986711798194432 (2011).
- 4 Neuffer, P. D. The Bioenergetics of Exercise. *Cold Spring Harb Perspect Med* 8, doi:10.1101/cshperspect.a029678 (2018).
- 5 Divakaruni, A. S. & Brand, M. D. The regulation and physiology of mitochondrial proton leak. *Physiology (Bethesda)* 26, 192-205, doi:10.1152/physiol.00046.2010 (2011).

For example, I miss thermodynamic or other physiological justifications of the irreversibility of the ETC. Surely many bacteria rely on reversal of this process to generate reducing power in the form of NAD(P)H during photosynthesis (called reversed electron flow)!

This is a very good point by the reviewer. Unfortunately, we lack information about redox metabolite levels in mitosis to reliably calculate if reverse ETC activity would be likely to take place. However, the net ETC activity (which we ultimately care and used in our model) is highly unlikely to be reversed, as we observe mitochondria hyperpolarize in mitosis even in the presence of oligomycin (Figures 3A and 3B), which can only be achieved through forward ETC activity. Nevertheless, we now acknowledge the possibility of reverse ETC activity and discuss this more in our supplemental section (starting on line 209).

I also miss the biological interpretation of capacitances, or an argument why it would be constant.

The reviewer is correct that we omitted detailed description about capacitances in the original manuscript. In brief, capacitance is determined by the surface area and material properties of mitochondrial inner membrane. In order for capacitance to be solely responsible for our findings, the

surface area and/or material of mitochondrial inner membrane would have to radically change following mitotic entry and revert back to original state following metaphase-anaphase transition. Such fast changes in the physical structure of mitochondria have not been reported and are unlikely to happen considering the short duration of mitosis. This is further supported by our new imaging experiments of mitochondrial networks in mitosis, which shows that on (at least on a larger scale) mitochondria do not undergo any radical remodeling during mitosis in our model system. Please see our supplemental section (starting on line 226) for full discussion.

There is actually a lot of “which we assumed” in the model. Ideally, any time you read this, you either expect a good argument/justification/reference, or a sensitivity analysis to show it does not affect the conclusions.

Thank you for this critical feedback. We have now improved our description to clearly distinguish between the components we used for developing the model, that is i) our experimental data ii) background information iii) assumptions, as well as iv) the sensitivity analysis for validating the model. These changes are within our supplemental section and highlighted for the reviewer to find.

Reviewer #3 (Remarks to the Author):

The authors show that in murine lymphocytic leukemia cell line L1210, as well as other murine cell lines, the mitochondrial membrane potential oscillates. More specifically, mitochondrial hyperpolarization occurs shortly after the onset of prophase, peaks at the transition from meta- to anaphase and ends before cytokinesis, as was determined relative to mitotic events with known timing. Furthermore, by use of chemical inhibitors, it is determined that the mitochondrial hyperpolarization and its spike-like behavior depend on CDK1-activity. The authors also use chemical inhibitors to show that mitochondrial ATP synthesis is required for cell growth, but not for mitotic entry and progression of mitosis. Finally, ATP synthesis rates during mitosis were derived through modelling and verified using oxygen consumption rate measurements. These efforts indicate that ATP synthesis rates drop during mitosis.

The paper draws on two highly timely tools: (i) highly sophisticated tools for single cell analyses, and (ii) integration of dynamic data into a mathematical model. While biology is full of state measurements, here the authors quantified something that is much harder to determine, namely a rate (i.e. the one of ATP synthesis), which they do even in a manner that is time resolved. This is by far not trivial and is a major achievement by itself. Overall, I feel that this work could become a contribution for Nature Communications, if the following concerns can be addressed:

We would like to thank the reviewer for the constructive, clear and exceptionally thorough feedback. We have now addressed the reviewer’s comments and we believe that this has significantly improved our manuscript. Please see details below.

Major concerns:

1. Is there a crucial control missing? The authors draw several conclusions by the use of inhibitors (cf. Fig. 2). These inhibitors were dissolved in DMSO or EtOH (line 373-374). Could it be that the observed effect is only due to the vehicle? Isn’t it necessary to show vehicle controls?

We thank the reviewer for pointing this out and our apologies for omitting the vehicle controls. We have now included the controls (revised Figures 2F and 2G) and this has not changed our conclusions.

In addition, as requested by reviewer #1, we have also added control experiments showing that CDK1 activity is indeed perturbed by our chemical treatments. We have now repeated our perturbations to CDK1 activity and monitored phosphorylation levels of CDK1 targets using MPM2 antibody. Our results section now states (starting on line 180) that “We validated that RO-3306 and BMS-265246 inhibit CDK1 activity by using western blotting with MPM2 antibody (Figure 2B), which identifies CDK1/2-phosphorylated sites found on various proteins^{38,39}. We also quantified the MPM2 antibody staining using flow cytometry (Figures 2C).”

2. Strange finding: expression of metabolic reporters completely alters behavior: The authors state that expression of exogenous metabolic FRET-based reporters alters mitotic behavior of the mitochondrial membrane potential. According to the authors, the strong CMV promoter used to express the reporters could cause the change in mitochondrial metabolism.

a) Why do the authors not use a weaker promoter to express these reporters? Why does such a strong expression of an exogenous protein lead to a completely different behavior? What kind of conclusions can be drawn from this for the system that they are studying?

b) More crucially, however, N-terminal tagging of geminin with monomeric Azami-Green (mAG) under control of the endogenous promoter also brings about a change in the mitochondrial hyperpolarization during mitosis (fig. S9B/C). Thus, it is unclear whether the changes in metabolism are caused by highly expressed exogenous proteins or by expression of fluorescent proteins in general.

The reviewer’s questions are indeed interesting and we have now clarified our conclusions in the main text. Our results section now states (starting on line 234) that “Since lower mitochondrial hyperpolarization in mitosis was observed with all genetic constructs tested, we attribute this change in hyperpolarization to increased protein expression rather than the specific protein that was expressed. Regardless of the mechanism affecting hyperpolarization, these results indicate that the genetically encoded metabolic reporter systems can bias quantitative analyses of mitotic mitochondrial bioenergetics in this model system.” While we are very interested in this phenomenon, we lack detailed evidence of the mechanism through which mitochondrial hyperpolarization is affected and thus we feel that any more discussion on this would only confuse the reader and take emphasis away from our main findings. We would also like to highlight that the work needed to uncover the mechanism through which mitochondrial hyperpolarization is affected is beyond the scope of this work.

Regarding the question “Why do the authors not use a weaker promoter to express these reporters?”, the reason is due to technical limitations in reliable signal detection. FRET based reporters suffer from low fluorescent signal intensities, weak signal-to-noise ratios and often require strong expression of the reporter constructs and long exposure times (~1s) to overcome these inherent limitations (*Leavesley, S. J. & Rich, T. C. Overcoming limitations of FRET measurements. Cytometry A, 2016. ; Algar, W. R., Hildebrandt, N., Vogel, S. S. & Medintz, I. L. FRET as a biomolecular research tool - understanding its potential while avoiding pitfalls. Nat Methods, 2019*). Unfortunately, our optical setup doesn’t allow reliable detection with lower expression of the reporters, especially since we are limited to brief (~100ms) exposure times. Our research questions also require repeated measurements which induce photobleaching, further increasing the need for a strong reporter signal. Thus, using our current measurements approach, we are unable to monitor mitotic bioenergetics with the FRET based reporters if they are expressed at lower levels. Furthermore, since the FUCCI cells with very low exogenous protein expression levels also displayed altered mitochondrial hyperpolarization in mitosis, it seems unlikely that one could obtain a cell line expressing the reporter constructs where mitochondrial behavior is completely unaffected, at least in our model systems.

3. Are model conclusions robust? The model is a central element on the basis of which they obtained the final conclusion of their paper namely that mitochondrial ATP synthesis rates drop during mitosis.

a. The authors assumed a binary state in their CDK activity (either on or off). Also, for their analyses, they assumed times when the CDK1 activity would either be on or off. It is not clear in how far their results would change when these times are changed. Or when the CDK1 activity would be considered to have more gradual dynamics. Would the outcome/conclusions change?

The reviewer brings up an important point. To check the robustness of the model, we performed sensitivity analyzes in the original manuscript. Our modeling approach has a separate time (Δt) which reflects duration of CDK1 on/off transitions. In our sensitivity analysis, we changed the Δt by 50% and observed that our final conclusion remains the same (i.e. ATP synthesis rates drop during mitosis at least 50%) (Figure S14). We do not apply our model to the Δt region, as this region contains too few datapoints for reliable fitting, as the Δt duration is significantly shorter than the CDK1_{on} region (Gavet, O. & Pines, J. *Progressive activation of CyclinB1-Cdk1 coordinates entry to mitosis. Dev Cell, 2010*). Instead, we have linearly interpolated the R_{ATP} values between CDK1_{on} and CDK1_{off} states to capture the gradual change in ATP synthesis. We still recognize the reviewer's criticism and we have now pointed this out in our main text (starting on line 254) that: "Furthermore, we accounted for the short duration when CDK1 activity is converting from one state to another³⁶ (Supplemental Note 5)", but we believe that the full details of this are too technical for the main text and therefore best left to the supplemental section.

b. Fig. 4F: it is not clear to me how the authors determine the ATP synthesis in G2 from the single-cell data, as they fit their model only to the transient $\Delta \psi$ dynamics, which is not in G2.

Again, we would like to thank the reviewer for bringing this up. Our modeling separates CDK1_{on} and CDK1_{off} regions. Although fitting was not carried out in G2, exactly as the reviewer states, we have extrapolated our RC values from the CDK1_{off} region to also cover G2. We believe this extrapolation is justified because our results show that mitochondrial membrane potential in G2 is comparable to the level at which membrane potentials return to in the fitted CDK1_{off} region (cytokinesis). We have now clarified this point in our results section (starting on line 266) by stating that: "We also utilized the CDK1_{off} state RC values to derive ATP synthesis in late G2." In addition, our figure legend for Figure 4F now states "G2 rates were obtained using RC values for CDK1 off and the $\Delta \Psi_m$ observed prior to mitotic entry."

c. Can one really compare the single-cell and the population-level experiments as shown in Fig. 4F?

The reviewer is correct that these approaches have some fundamental differences and thus quantitative comparisons may not be accurate. However, the population-based method is considered as 'golden standard' (oxygen consumption measurements with and without oligomycin) for quantifying mitochondrial ATP synthesis rates, and we believe that it is crucial to compare our modeling results to the existing standard methods. In addition, carrying out both measurements further validate our main discovery that mitochondrial ATP synthesis in mitosis is decreased. However, we recognize the reviewer's point and we have now moved these results in to separate figures (the population-level measurement is now shown in Figure 5B) to avoid misleading comparisons.

d. Can the modelling results shown in fig. 4G be verified experimentally, e.g. with another inhibitor that leads to an anaphase block (potentially proTAME)?

We have carefully considered the reviewer's suggestion. If we understood the reviewer's point correctly, we are asked to validate our model's final conclusion (ATP synthesis is inhibited in mitosis) using alternative TMRE monitoring data where cells are arrested in mitosis. We cannot derive the result shown in Fig. 4G using any mitotic inhibitors, as our modeling requires data from both early (CDK1_{on}) and late (CDK1_{off}) mitosis. However, we can validate that the conclusions we draw about RC values in early mitosis (CDK1_{on}) are not biased due to the short duration of early mitosis which provides us with relatively low amount of datapoints for model fitting. We have now compared the RC values we obtain from normal mitosis (same data as in Fig. 4) to RC values we obtain when cells are arrested in mitosis with STLC. While in normal mitosis our fitting is done on ~15min data segment, in STLC-treated cells that arrest in mitosis we can carry out the fitting over much longer durations (~2h). We have now compared control and STLC-treated cells (with or without additional treatment with oligomycin) and we found that the RC values (time constants) are not statistically different from each other (Figure R3, below). Thus, our modeling is not biased by the short fitting region during early mitosis. However, since this comparison is technical in nature and doesn't provide any additional conclusions on the biology we're studying, we have decided not to include this result in our manuscript.

Figure R3. Time constants (RCs) extracted using our model from control samples (normal mitosis) and from STLC treated samples (prometaphase arrest) during early mitosis, where CDK1 is active. Data is shown for both untreated and oligomycin treated cells. In both cases our model derives the same RC values whether cells go through normal mitosis or arrest in prometaphase.

4. Dynamics of ATP levels is a crucial, but lacking piece of information in the work: A key element that is missing in the story are data on ATP levels before, during and after mitosis. The reason for this is the following: On the one hand, the authors state in the discussion that the turnover rate of the ATP is in the minute range, meaning that if the synthesis of ATP would be stopped, ATP would drop to zero in a few minutes. In the discussion, the authors state that in mitosis there is significant ATP consumption (i.e. they stick to the current notion that mitosis requires lots of ATP). In their work, they claim that ATP synthesis drops in mitosis. Taking these 3 pieces of information together, this would mean that ATP levels NEED to drop during mitosis, if they are right with their finding / if the current notions is correct. If a measurement of ATP levels during mitosis would not show any drop, this would mean that either their finding is not correct, or that the current notion (i.e. mitosis requiring lots of ATP) is incorrect. In any case, this would advance our understanding tremendously. Notably, while many papers working on energy homeostasis measure metabolite level dynamics, and then to guess-work on the rates of the underlying processes, here the authors did the much harder thing (i.e. they quantified a rate), but did not look at the dynamics of the state variable (i.e. ATP) itself. I feel that the authors should measure in synchronized cultures (potentially by means of metabolomics) the levels of ATP during the cell cycle. This would make the "package" complete.

First, we would like to thank the reviewer for acknowledging that our approach does something rare and difficult by measuring the rate of ATP synthesis instead of only looking at ATP levels. We also fully agree with the reviewer that the ATP levels need to be assessed as well. We have now measured both ATP and ADP levels using a standard luciferase-based biochemical assay in cells synchronized to G2 and mitosis. This revealed that ATP levels decrease ~40% in mitosis and ADP levels decrease approximately equally. As we already pointed out in our manuscript, decreased ATP levels in mitosis have also been observed in previous literature. We now discuss these findings and their implications (impact on, for example, AMPK activation, allosteric regulation of enzymes and phase separation of disordered proteins) in our updated discussion. Please see our results section (starting on line 297), Figure 5D and our discussion (starting on line 357) for full details.

Notably, as suggested by reviewers #1, #2 and #4, we have now also measured glycolytic rates by comparing lactate efflux in cells in G2 and in mitosis. Our results show that mitotic cells have ~50% lower lactate efflux rate, suggesting decreased glycolytic ATP synthesis rates in mitosis. This, together with our measurements of mitochondrial ATP synthesis rates, argues that the overall ATP synthesis is lower in mitosis than in G2. As cellular ATP pools are fully consumed in ~1 min in interphase, ATP consumption needs to decrease following the overall decrease in mitotic ATP synthesis. Thus, these results support our original conclusion that cell division is not highly energy consuming in comparison to interphase. Please see our results section (starting on line 293), Figure 5C and our discussion (starting on line 329) for full details.

Minor concerns:

1. I find the discussion section weak and not adequate for a high profile paper: What are the main advances? What are the key aspects that will require further investigations? What are the key implications of the work?

We thank the reviewer for bringing this up. As this was a key concern for several reviewers, we have now significantly expanded our discussion sections to summarize, emphasize and discuss key implications. We believe that this has significantly improved our manuscript.

2. For some key data, only data from an individual cell is shown, e.g. Fig. 1B, C and D. Is it possible to show data from more cells (or more cell cycles); ideally like the cell trace shown in Fig. 1A. These are key figures and I would not feel comfortable only having seen data from an individual cell from and individual cycle.

Thank you for pointing this out. We now include more experimental replicates in the supplemental section (Figures S4A, S4B and S5).

3. Comparing fig. 2B and E, I notice a difference between the 2 STLC+RO-3306 conditions in TMRE/mass signal at equilibrium. When STLC and RO-3306 are added simultaneously, the TMRE/mass signal at equilibrium is higher than during G2, whereas if RO-3306 is added only after the STLC treatment equilibrium has been achieved, the TMRE/mass signal drops back to (or below) the G2 state. How can this difference be explained?

This is simply due to the concentration of RO3306 used. When adding STLC and RO3306 together, we had to use a low concentration of RO3306 (partial inhibition of CDK1) as otherwise the cells will not be able to enter mitosis. However, after cells enter mitosis, we can completely reverse the TMRE increase by using higher concentration of RO3306. We have now clarified our figure legends regarding this. In addition, as requested by reviewer #1, we have carried out validation experiments of CDK1 activity (Figures 2B and 2C).

4. Page 9, line 170ff: This experiment assumes that the arrest does not affect the metabolic rates. Can this assumption be safely made? What if it is not true?

The reviewer raises an important point. While we aimed to minimize the cell cycle arrest durations, we cannot prove that all metabolic rates remain unaffected by cell cycle arrest. However, as shown in Figure 1D and in our preprint work (Mu et al., Mass measurements of polyploid lymphocytes reveal that growth is not size limited but depends strongly on cell cycle, BioXriv, <https://doi.org/10.1101/2019.12.17.879080>), cell cycle arrests in G2 do not interfere with cell growth rates, suggesting that G2 arrest is not radically affecting cell metabolism. Nonetheless, it is possible that even short mitotic arrests can influence metabolic rates, and this might be reflected in our data. In fact, this could explain why our modeling approach suggests slightly higher mitochondrial ATP synthesis rates in mitosis than the more classical population-based oxygen consumption measurements which require cell cycle arrests. We view the ability to monitor unsynchronized single cells as a major strength of our approach. We have now amended our writing by clearly stating that population based measurements were carried out on synchronized cells and we also state in the figure legend for population-based oxygen consumption measurements (Figure 5B) that “Note that the result is similar to that obtained with single-cell modeling of $\Delta\Psi_m$ dynamics (Figure 4F), but the modeling approach does not rely on cell cycle synchronizations.”

5. Goodness of fit (R^2 ; model to experimental data) varies for each single cell trace (fig. S11A). Is there a general indication of R^2 to be given?

Yes, we have now added model fitting R^2 values (histograms of single-cell values & mean values, Figure S12D). For all sample groups (control and oligomycin, $CDK1_{on}$ and $CDK1_{off}$ fittings) the average R^2 values > 0.9 .

Suggestions:

1. fig. 2B/C: change order

We have now significantly changed our Figure 2 and these changes include a change of figure order.

2. fig. 3A: show typical no treatment cell traces to make figure interpretation more intuitive.

Thank you for the suggestion. We've now done this.

3. fig. 3C/D: extend the x-axis to $t = -15$ min to show start of oligomycin treatment.

Thank you for the suggestion. However, we did not systematically collect samples at $t = -15$ min, so we feel that the plot would become more confusing by changing the axis without showing $t = -15$ min data.

4. fig. 3E: change the y-axis label of the boxplot to Growth rate $t = 1$ h .

Done, thank you for the suggestion.

5. Page 7: lines 110f: Mention that the oxygen consumption measurements were done on synchronized cells. This will help the reader.

We have now clarified the text as the reviewer suggested.

6. Fig. 4G: I find “ATP synthesis amount” (also somewhere mentioned in the text) a strange term

We've now changed the expression to “total ATP synthesized”.

7. Fig. 4F: what are the n's indicated for the OCR experiments? Are these independent synchronizations and measurements?

We apologize for the lack of clarity in the figure legends. We have now clarified that these are “independent synchronizations and cultures”.

8. Page 9, line 175: leakage of what?

Thank you for spotting this. We have now corrected the writing to say “proton leakage”.

9. Page 5, line 100: “we observed that PARTIAL CDK1 inhibition...”

Thank you for spotting this. We have now corrected the writing.

10. In figure 1, “non-quenching” and “quenching” concentrations are highlighted. For a reader not particularly familiar with the used dyes, this might be difficult to grasp. Thus, to enhance accessibility for a broader readership, I suggest that the author explain this, also with a short comment in the main text.

We fully agree with the reviewer. Our revised manuscript explains this and other experimental setups in more detail.

11. The authors find that certain mouse lymphocytes do not show the characteristic spike in $\Delta\Psi_m$ increase. Could they speculate why this is the case, and whether this provides any further insight for their story? Would these cells have hardly any respiratory activity to start with?

This is indeed an interesting question. We have examined growth, metabolic and RNA-seq data from the FL5.12 cells in comparison to the L1210 cell line (data collected in previous manuscripts by the Manalis lab, please see <http://manalis-lab.mit.edu/publications.html> for publications). We have not found a clear reason why the FL5.12 cells do not display the increased $\Delta\Psi_m$ in mitosis. We feel that in the absence of clear mechanistic insight this would only confuse the reader and have therefore omitted all discussion on this.

Corrections:

1. Line 137: analyses (instead of analyzes)
2. Line 395: G1/S cells are described as having >4N DNA content, this should be <4N.
3. Page 5, line 96: Reference to figure missing
4. Fig. 2B: vertical dashed line missing
5. Fig. 2B: legend in the figure and the caption text do not match with regards to the colors mentioned
6. Page 11, line 199: Reference needed for the K_m values being in the micromolar range. (I am in fact not certain whether this statement (without reference) is really correct).
7. Fig. 3C/D: Specify what the control cells were

Thank you for spotting these errors. We have now corrected all of them accordingly.

We would like to specifically note the point #6 about K_m values being in micromolar range. While we have now added a reference for this, we have also realized that this is an assumption that derives from *in vitro* enzymatic assays and does not necessarily cover all enzymes/kinases. Thus, we have also changed our writing to reflect this uncertainty. We now state (starting on line 357) that “most enzymes in the cellular environment are believed to have Michaelis constants (K_m) for ATP in the micromolar range⁵⁵.”.

Reviewer #4 (Remarks to the Author):

The authors seek to quantify two mitochondrial parameters during mitosis: 1) membrane potential and 2) rate of mito ATP synthesis.

For #1, they apply a neat method to measure mass of cells to then compute mass-normalized mito membrane potential at high temporal resolution in asynchronous, normally dividing single cells. They report acute mitochondrial hyperpolarization during mitosis, with a peak at the metaphase-anaphase transition. These data need further support and refined interpretation:

- 1) The current experiments rely on whole-cell TMRM fluorescence. The authors should sort the cell

populations of interest and measure membrane potential in permeabilized cells and in whole cells by microscopy for finer spatial resolution. Although permeabilization is artificial, it may help reinforce their claims and overcome confounders such as mitotic swelling and plasma membrane hyperpolarization.

We thank the reviewer for suggesting this. We have carefully considered such experiments, but we believe such experiments would be difficult or even impossible to execute and interpret due to following reasons: 1) While sorting of mitotic cells can be done, the short duration of mitosis results in a situation where the sorted cells would have already exited mitosis before any imaging could be done. 2) We are not aware of the signaling & metabolic state required to maintain mitochondria hyperpolarized in mitosis. Thus, permeabilization and the accompanied change in cytosolic composition could very easily result in a situation where mitochondrial hyperpolarization changes due to the permeabilization. Consequently, we would have no way to distinguish between false negative and real negative results or between false positive and real positive results.

We still acknowledge the reviewer's point about validating our results with finer spatial resolution. To this end, we have now imaged mitochondria stained with TMRE in live L1210 cells and observed that the intracellular TMRE distribution between mitosis and interphase seems identical (Figures 1H and S7A). In other words, TMRE staining retains mitochondrial specificity in mitosis, thus supporting our original conclusion that TMRE is a reliable proxy for $\Delta\Psi_m$ in our model system. In addition, during mitosis mitochondria do not display radical variability in the level of TMRE staining, suggesting that the mitochondrial hyperpolarization is relatively uniform within each cell. We report these results in our manuscript (starting on line 146) and we conclude that: "Furthermore, we did not observe differences in the intracellular distribution of the TMRE signal between mitotic and interphase cells (Figures 1H, Figure S7A), suggesting that mitochondria hyperpolarize uniformly in mitosis, and that the TMRE signal increase in mitosis is not due to increased cytosolic TMRE."

In addition, we have now carried out a more classical comparison between $\Delta\Psi_m$ and cell cycle stage, by comparing how MitoTracker Red CMXRos ($\Delta\Psi_m$ sensitive probe) signal differs between mitotic and G2 cells. Mitotic and G2 cells were separated using DNA content, Histone H3 (Ser10) phosphorylation and Cyclin-B levels. Consistent with our previous analyzes, this revealed that mitotic cells have higher levels of $\Delta\Psi_m$. Please see our results section (starting on line 133) and Figures S5E,F. Although these experiments lack intercellular spatial resolution of $\Delta\Psi_m$, they provide a completely independent approach to support our original conclusion about $\Delta\Psi_m$ being higher in mitosis than in G2. Please also note that our modeling of ATP synthesis rates does not require finer spatial resolution.

2) The authors seem to be wholly unaware of the landmark work of Toren Finkel and Lippincott-Schwartz — both prominent in this field — that have written key papers within the past decade on the coordination of mitochondrial bioenergetics with the cell cycle. Are the authors unaware of this work which is foundational for the current paper?!

Thank you for pointing this out. We have not been unaware of the many elegant studies done by Finkel and Lippincott-Schwartz labs, however, in our original short format manuscript we did not see the need to discuss their work extensively as they have mainly focused on mitochondrial morphology, not ATP synthesis. However, we fully acknowledge the reviewer's criticism and we have now significantly extended our introduction section to discuss the background work done by these labs and others.

3) Informed with this literature, the authors must perform microscopy analysis — using TMRM staining and mitochondrial markers as a function of time — to determine whether the membrane potential signal is simply an artifact of mitochondrial fusion and fission.

Here again the reviewer has a very good suggestion that is unfortunately not technically feasible. We have tried time-lapse imaging of live L1210 cells stained with TMRE, but unfortunately TMRE photobleaching is too significant to allow any proper analysis on such data. This may seem

surprising, but we would like to remind the reviewer that these are suspension cells and imaging the whole cell requires ~70 slices to be imaged (assuming a typical z-resolution of 0.2 μ m, the cells can be over 13 μ m tall). In fact, even in a single time point of TMRE imaging across the whole cell results in photobleaching that makes quantitative analysis challenging (Figure R4, below).

Despite not being able to carry out time-lapse imaging, as the reviewer suggested, we have partially addressed the reviewer's point by investigating how mitochondrial networks change in mitosis in our model system. Using mitochondrial RFP expressing L1210 cells, we have quantified mitochondrial network changes between mitosis and interphase, observing that our model system displays only modest mitochondrial fission in mitosis. Starting on line 154 in our results section we now say: "To analyze the mitochondrial morphology independently of $\Delta\Psi_m$, we expressed mitochondrially localized RFP in L1210 cells. In mitosis the average lengths of mitochondria were lower than in interphase and, consistently, there were more individual mitochondria in mitotic cells than in interphase cells (Figures S7B-D). However, this mitochondrial fragmentation in mitosis was not extensive, as even in mitosis mitochondria remained mostly connected and only a few mitochondria covered most of the mitochondrial volume. While the mitotic increase in $\Delta\Psi_m$ could still be linked to mitochondrial fission in early mitosis, the decrease in $\Delta\Psi_m$ during cytokinesis does not correlate with the known mitochondrial fusion/fission changes¹⁴⁻¹⁶."

Last, but not least, we would like to clarify one very important point. The validity of our conclusion, that $\Delta\Psi_m$ increases and the rate of mitochondrial ATP synthesis decreases in mitosis, is not affected by the mechanism of $\Delta\Psi_m$ increase in any ways. In other words, our modelling approach and results will not change even if the $\Delta\Psi_m$ increase was due to changes in mitochondrial fusion and fission. We have now clarified this in our discussion section (starting on line 314) by stating that: "...the inhibition of ATP synthesis could also be influenced by, for example, changes in mitochondrial morphology. Importantly, the mechanism through which ATP synthase is inhibited does not influence the conclusions of our electrical circuit modeling."

Figure R4. Whole cell projections of microscopy images of a L1210 cell stained with TMRE. Left displays a projection from the top. Right displays a projection from the side. Image collection was started from the top of the cell. The photobleaching of TMRE signal towards the bottom of the cell is clear in the side projection.

4) There are two cases where the authors do not observe said mitotic mitochondrial hyperpolarization i) mouse FL5.12 lymphocytes and ii) cells over-expressing mitochondrial reporter constructs. The authors should try to explain these notable exceptions.

This is indeed an interesting question. We have examined growth, metabolic and RNA-seq data from the FL5.12 cells in comparison to the L1210 cell line (data collected in previous manuscripts by the Manalis lab, please see <http://manalis-lab.mit.edu/publications.html> for publications). We have not found a clear reason why the FL5.12 cells do not display the increased $\Delta\Psi_m$ in mitosis. We feel that in the absence of clear mechanistic insight this would only confuse the reader and have therefore omitted all discussion on this.

Regarding the cells expressing mitochondrial reporter constructs, we have now clarified our conclusions in the main text. Our results section now states (starting on line 234) that “Since lower mitochondrial hyperpolarization in mitosis was observed with all genetic constructs tested, we attribute this change in hyperpolarization to increased protein expression rather than the specific protein that was expressed. Regardless of the mechanism affecting hyperpolarization, these results indicate that the genetically encoded metabolic reporter systems can bias quantitative analyses of mitotic mitochondrial bioenergetics in this model system.” While we are very interested in this phenomenon, we lack detailed evidence of the mechanism through which mitochondrial hyperpolarization is affected and thus we feel that any more discussion on this would only confuse the reader and take emphasis away from our main findings. We would also like to highlight that the work needed to uncover the mechanism through which mitochondrial hyperpolarization is affected is beyond the scope of this work.

For #2, the authors model the mitochondrion as a circuit and derive the rate of mitochondrial ATP synthesis. Their analytical and empirical data from the circuit-based approach is supported by supplementary traditional OCR measurements. Altogether, the authors report reduced mitochondrial ATP synthesis. However, extending this observation to “challenge the traditional dogma that cell division is a highly energy demanding process.” (abstract) and “suggests a much lower rate of ATP consumption during mitosis than previously assumed” (discussion) is premature as it wholly ignores ATP demand and it wholly ignores glycolytic ATP synthesis. For clarity -- if the cell is not using ATP (no load on the resistor), there is no need for mito ATP production. For clarity -- if the cell is doing plenty of glycolysis, there will be plenty of glycolytic ATP and the cells will not need to do mito ATP OXPHOS. Recall respiratory control, mito respiration is primarily regulated by ADP availability.

We completely agree with the reviewer and we have now investigated glycolytic rates by comparing lactate efflux in cells in G2 and in mitosis (as also suggested by other reviewers). Our results show that mitotic cells have ~50% lower lactate efflux rate, suggesting decreased glycolytic ATP synthesis rates in mitosis. This, together with our measurements of mitochondrial ATP synthesis rates, argues that the overall ATP synthesis is lower in mitosis than in G2. As cellular ATP pools are fully consumed in ~1 min in interphase, ATP consumption needs to be decreased following the overall decrease in mitotic ATP synthesis. Thus, these results support our original conclusion that cell division is not highly energy consuming in comparison to interphase. Please see our results section (starting on line 293), Figure 5C and our discussion (starting on line 329) for full details.

We would also like to thank the reviewer for reminding us about the role of ADP in respiratory control, as this is indeed critical for interpreting our results. We have now measured ATP and ADP levels using a standard luciferase-based biochemical assay in cells synchronized to G2 and mitosis. This revealed that ATP levels decrease ~40% in mitosis and ADP levels decrease approximately equally. As we already pointed out in our manuscript, decreased ATP levels in mitosis have also been observed in previous literature. We now discuss these findings and their implications (impact on, for example, AMPK activation, allosteric regulation of enzymes and phase separation of disordered proteins) in our updated discussion. Please see our results section (starting on line 297), Figure 5D and our discussion (starting on line 357) for full details.

We feel that these measurements have significantly improved our manuscript’s impact and we would like to thank the reviewer for this very constructive feedback.

REVIEWER COMMENTS

Reviewer #1 (Remarks to the Author):

The manuscript has been substantially improved. The authors have addressed most of my critics and suggestions. My only suggestion is to specify in the title that this model is valid in the studied cells.

Reviewer #2 (Remarks to the Author):

I am generally happy with the changes made to the manuscript, and the answers to my questions.

They only misunderstood my comment on the reversibility of the ETC: I did not mean to suggest that the flux would actually be reversed in mammalian cells, but rather, that the ETC is not irreversible and thus in principle sensitive to product inhibition. The question then is if this affects the conclusions.

In general, one would want to test a model's uncertainties by parameter sensitivity analysis or alternative model schemes, something the authors have now done also in response to reviewer 3.

Reviewer #4 (Remarks to the Author):

The authors have done a good job of ensuring that the TMRM signal is not an optical artifact related to fusion / fission status during mitosis. And the authors have also been responsive to my previous suggestion of measuring ATP-coupled OCR, ATP-coupled glycolysis, ATP levels during mitosis. -- this new data presented in Figure 5 is sufficient to support the title claim that ATP production declines during mitosis. However, I continue to have major concerns:

(1). Scholarship: In the opening paragraph, the authors set up the claim that the major "dogma" is that mitosis "requires large amounts of energy." Clearly this claim helps to elevate the perceived importance of the work. But on what literature / factual basis are they making this claim? For example, here is a nice figure from Ron Vale and Jonathan Weissman that shows that protein translation goes down in mitosis:

<https://elifesciences.org/articles/07957/figures>

So what is this evidence for this major "dogma?" I would encourage the authors to review the primary literature and then provide a balanced, scholarly view (including citing the above work). References in 1-7 appear to be related to autophagy, mitophagy, calcium. I am surprised that the authors claimed to have been very familiar with the work of Lippincott Schwartz and Toren Finkel, yet chose not to cite their work in the initial submission.

(2). Cell-type and cell-state specificity: The principle experimental observation in the paper is that the membrane potential rises during mitosis using a very neat technique. But this is clearly a highly cell type specific phenomenon (FL5.12 lymphocytes don't do this) and cell state dependent (expressing a mitochondrial fluorescent protein abrogates it). I had asked the authors to explore this mechanism, which they have not. At the very least they ought to need to caveat the results — even in the abstract — that this there is an increase in membrane potential "in some cell types and cell states."

(3). Contrived use of the mathematical model: The authors introduce and utilize a complex circuit model to interpret the results of the hyperpolarization. First, Nicholls and others have introduced very simple circuit models and they are often used to interpret results from bioenergetics studies. Is a new model really needed here? In Figure 5, they show that ATP-coupled OCR goes down; glycolysis goes down; steady state ATP levels go down — the most parsimonious explanation is that ATP production is

down — and these measurements are sufficient to justify the title claim. If the model is so useful, can they tell us why FL5.12 don't do this and why expression of FUCCI abrogates the hyperpolarization)? The paper may actually be clearer and more rigorous if the model is eliminated and simple inference from Figure 5 is made.

Reviewer reply letter by Kang, Katsikis, et al.

Dear reviewers and editors of Nature Communications, thank you for reviewing our work. We've now addressed the reviewer comments and you can find the comments below (in black) along with our point-by-point responses (in blue). In our attached manuscript, all relevant changes are highlighted in grey.

Point-by-point replies to reviewer comments

Reviewer #1 (Remarks to the Author):

The manuscript has been substantially improved. The authors have addressed most of my critics and suggestions. My only suggestion is to specify in the title that this model is valid in the studied cells.

We thank the reviewer for the feedback. We have now changed our title to specify the model system used. In addition, we have clarified in our abstract that “We observe similar mitochondrial hyperpolarization in primary T cells, but not in all cell lines tested.”

Reviewer #2 (Remarks to the Author):

I am generally happy with the changes made to the manuscript, and the answers to my questions.

They only misunderstood my comment on the reversibility of the ETC: I did not mean to suggest that the flux would actually be reversed in mammalian cells, but rather, that the ETC is not irreversible and thus in principle sensitive to product inhibition. The question then is if this affects the conclusions.

In general, one would want to test a model's uncertainties by parameter sensitivity analysis or alternative model schemes, something the authors have now done also in response to reviewer 3.

We thank the reviewer for the feedback and we apologize for misunderstanding the reviewer's comment. As the reviewer pointed out, we have tested the model uncertainties both experimentally and by carrying out sensitivity analyses. Our model doesn't directly include ETC regulation by product inhibition, i.e. by high mitochondrial membrane potential. However, as detailed in our previous response regarding the reversibility of the ETC, we have experimentally validated that ETC flux direction is unchanged even in the presence of oligomycin. This does not exclude the possibility of a minor change in ETC activity due to product inhibition, but this is unlikely to influence our results in a meaningful way, because i) we have validated our results against independent measurements of ATP synthesis (oxygen consumption assays), and ii) our model compares the CDK_{1on} and CDK_{1off} regions of the same cell, which minimizes systematic biases, such as ETC activity changes due to product inhibition. We also note that product inhibition of ETC activity is considered small when comparing control and oligomycin treated cells in oxygen consumption assays (Brand, M.D. & Nicholls, D.G. Assessing mitochondrial dysfunction in cells. *Biochem J*, 2011).

Reviewer #4 (Remarks to the Author):

The authors have done a good job of ensuring that the TMRM signal is not an optical artifact related to fusion / fission status during mitosis. And the authors have also been responsive to my previous

suggestion of measuring ATP-coupled OCR, ATP-coupled glycolysis, ATP levels during mitosis. -- this new data presented in Figure 5 is sufficient to support the title claim that ATP production declines during mitosis.

We thank the reviewer for acknowledging the positive aspects of our work and we welcome the additional suggestions to improve our manuscript.

However, I continue to have major concerns:

(1). Scholarship: In the opening paragraph, the authors set up the claim that the major “dogma” is that mitosis “requires large amounts of energy.” Clearly this claim helps to elevate the perceived importance of the work. But on what literature / factual basis are they making this claim? For example, here is a nice figure from Ron Vale and Jonathan Weissman that shows that protein translation goes down in mitosis: <https://elifesciences.org/articles/07957/figures>

So what is this evidence for this major “dogma?” I would encourage the authors to review the primary literature and then provide a balanced, scholarly view (including citing the above work). References in 1-7 appear to be related to autophagy, mitophagy, calcium. I am surprised that the authors claimed to have been very familiar with the work of Lippincott Schwartz and Toren Finkel, yet chose not to cite their work in the initial submission.

We fully agree with the reviewer that there is no proper evidence for how ATP synthesis or consumption change in mitosis and the common assumption that mitosis consumes a lot of energy is not experimentally verified. We have pointed this out in our manuscript and, thanks to the reviewer’s feedback, we have now improved our writing to highlight this even more (starting on lines 3, 20, 57 and 345). We have also removed the word ‘dogma’ from our abstract and introduction, as we suspect this word is partly the source of confusion, and instead we simply state that “our results ... suggest that cell division is not a highly energy demanding process”. We have also cited the manuscript by Vale and Weissman, while expanding our discussion section about protein synthesis and other energy consuming processes in mitosis (starting on line 371). Finally, we have clarified to the reader that the exact levels of protein synthesis in mitosis have remained controversial in the literature, possibly due to cell cycle synchronization induced artefacts and cell type dependent differences, but in the L1210 cells protein synthesis remains highly active in mitosis.

(2). Cell-type and cell-state specificity: The principle experimental observation in the paper is that the membrane potential rises during mitosis using a very neat technique. But this is clearly a highly cell type specific phenomenon (FL5.12 lymphocytes don’t do this) and cell state dependent (expressing a mitochondrial fluorescent protein abrogates it). I had asked the authors to explore this mechanism, which they have not. At the very least they ought to need to caveat the results — even in the abstract — that this there is an increase in membrane potential “in some cell types and cell states.”

We fully agree with the reviewer that our main conclusions are limited to one model system and we want to avoid any overstatements about how general this cell behavior is. Following the reviewer’s suggestion, we now specify the model system used in the title. In addition, we have clarified in our abstract that our results are observed “when cells are in minimally perturbed state.” and that “We observe similar mitochondrial hyperpolarization in primary T cells, but not in all cell lines tested”. We have now also specified in our results (starting on line 238) and discussion (starting on line 313) sections that the mitochondrial hyperpolarization is not observed in all cell types tested and that cell state and excessive protein expression can influence the mitochondrial hyperpolarization. Similar comment was also made by reviewer #1 and we would like to thank both reviewers for pointing this out.

(3). Contrived use of the mathematical model: The authors introduce and utilize a complex circuit model to interpret the results of the hyperpolarization. First, Nicholls and others have introduced very simple circuit models and they are often used to interpret results from bioenergetics studies.

Our electrical circuit model is directly based on the models published by Nicholls and others, and is similar in complexity (see, for example, Fig. 1B from Brand, M.D. & Nicholls, D.G. Assessing mitochondrial dysfunction in cells. *Biochem J*, 2011). We recognize that this may have been unclear before and we now clarify this to the reader (lines 246 and 251). However, our quantitative use of the model (fitting experimental results to its analytical solution to derive quantitative conclusions about the ATP synthesis rate) may have resulted in additional complexity in the presentation. We have now significantly simplified our writing in our results section (on lines 245-265) to clarify the presentation of our model.

If the model is so useful, can they tell us why FL5.12 don't do this and why expression of Fucci abrogates the hyperpolarization)?

We acknowledge that our model doesn't have the complexity and depth to explain cell type specific differences in metabolism. Our modelling is simply aimed at deriving quantitative information about the dynamics of mitochondrial ATP synthesis using the single-cell $\Delta\psi_m$ monitoring data that we gather. We now directly acknowledge this in our results section when we introduce the model (starting on line 245). The reviewer's question about cell type- and state-dependent changes in mitochondrial metabolism is very interesting and, although we cannot answer it in our current work, we have acknowledged this limitation of our manuscript, as discussed above under reviewer's point #2.

Is a new model really needed here? In Figure 5, they show that ATP-coupled OCR goes down; glycolysis goes down; steady state ATP levels go down — the most parsimonious explanation is that ATP production is down — and these measurements are sufficient to justify the title claim. The paper may actually be clearer and more rigorous if the model is eliminated and simple inference from Figure 5 is made.

The reviewer is correct that our results shown in Fig. 5 alone would lead to the same main conclusion (ATP synthesis is reduced in mitosis) even if we did not have our modeling results. However, our modeling approach substantiates the conclusions regarding the mitochondrial ATP synthesis, and provides additional information that would not be otherwise available. First, our modeling approach results in similar conclusions about mitochondrial ATP synthesis rates in mitosis as population-based oxygen consumption assays, despite the fact that these methods are fully independent. Second, cell cycle synchronizations can perturb metabolism, and thus the results of any population-based metabolic measurements in synchronized cells may be biased. In contrast, our modeling approach allows us to monitor mitotic ATP synthesis in the absence of any cell synchronization. Third, our modeling approach also reveals the dynamics of mitochondrial ATP synthesis rate during mitosis, which are required for calculating of the overall decrease in the amount of ATP synthesized in mitosis (Fig. 4g). Such information cannot be derived from population-based oxygen consumption measurements due to lack of temporal resolution (i.e. no information about the dynamics).

We now realize that our writing did not properly motivate our use of the model. To address this, we have now added a short paragraph to our discussion to point out these differences (starting on line 335). We thank the reviewer for highlighting this issue.